# SKA2 regulated hyperactive secretory autophagy drives neuroinflammation-induced neurodegeneration

Jakob Hartmann [1] ✉, Thomas Bajaj[2], Joy Otten[1,3], Claudia Klengel[1], Tim Ebert[2], Anne-Kathrin Gellner[4], Ellen Junglas[2], Kathrin Hafner[5], Elmira A. Anderzhanova[2,6], Fiona Tang[1,7], Galen Missig[1], Lindsay Rexrode[8], Daniel T. Trussell [8], Katelyn X. Li[1], Max L. Pöhlmann[1,7], Sarah Mackert [2], Thomas M. Geiger[9], Daniel E. Heinz [2,6], Roy Lardenoije[1,3], Nina Dedic[1], Kenneth M. McCullough [1], Tomasz Próchnicki[10], Thomas Rhomberg [1], Silvia Martinelli [5], Antony Payton [11], Andrew C. Robinson [12,13], Valentin Stein [14], Eicke Latz [10,15], William A. Carlezon Jr[1], Felix Hausch[9], Mathias V. Schmidt [7], Chris Murgatroyd [16], Sabina Berretta[1], Torsten Klengel [1,3], Harry Pantazopoulos[8], Kerry J. Ressler [1,17] ✉ & Nils C. Gassen [2,5,17] ✉

High levels of proinflammatory cytokines induce neurotoxicity and catalyze inflammation-driven neurodegeneration, but the specific release mechanisms from microglia remain elusive. Here we show that secretory autophagy (SA), a non-lytic modality of autophagy for secretion of vesicular cargo, regulates neuroinflammation-mediated neurodegeneration via SKA2 and FKBP5 signaling. SKA2 inhibits SA-dependent IL-1β release by counteracting FKBP5 function. Hippocampal *Ska2* knockdown in male mice hyperactivates SA resulting in neuroinflammation, subsequent neurodegeneration and complete hippocampal atrophy within six weeks. The hyperactivation of SA increases IL-1β release, contributing to an inflammatory feed-forward vicious cycle including NLRP3-inflammasome activation and Gasdermin D-mediated neurotoxicity, which ultimately drives neurodegeneration. Results from protein expression and co-immunoprecipitation analyses of male and female postmortem human brains demonstrate that SA is hyperactivated in Alzheimer's disease. Overall, our findings suggest that SKA2-regulated, hyperactive SA facilitates neuroinflammation and is linked to Alzheimer's disease, providing mechanistic insight into the biology of neuroinflammation.

Microglia, the resident immune cells of the brain, have critical roles in tissue homeostasis, phagocytic activity and cytokine production. Increasing amounts of pro-inflammatory cytokines, such as IL-1β can be harmful and toxic to neurons and have been associated with neurodegenerative illnesses including Alzheimer's disease (AD)[1-4].

However, the specific release mechanisms of pro-inflammatory cytokines from microglia that govern neuroinflammation-driven neurodegeneration are not fully understood.

Secretory autophagy (SA), a non-lytic modality of autophagy for secretion of vesicular cargo involving a stepwise succession of

autophagy signaling proteins, cargo receptors and RQ-SNARE fusion proteins, has been linked to peripheral immune responses and inflammation[5,6]. However, the key molecular mechanisms involved remain elusive and it is unknown whether SA may play a role in neuroinflammation.

Recently, we identified the stress-inducible co-chaperone FK506-binding protein 51 (FKBP5) as a scaffolding protein and key driver of SA that facilitates fusion of the secretory autophagosome with the plasma membrane and subsequent cargo secretion to the extracellular milieu[7]. FKBP5 promotes the RQ-SNARE complex formation between the secretory autophagosome and the plasma membrane through interaction with several of its key components including vesicle-trafficking protein SEC22B and synaptosomal-associated protein 29 (SNAP29). Interestingly, a scaffolding protein, spindle and kinetochore-associated complex subunit 2 (SKA2), has previously been identified as a potential binding partner of SNAP29 in cervical adenocarcinoma (HeLa S3) cells[8], leaving the role of SKA2 in the brain, and its potential involvement in SA, unexplored.

SKA2 is primarily known to be part of the spindle and kinetochore (SKA) complex, which also includes SKA1 and SKA3. This complex plays a vital role in the kinetochore-microtubule interface and is necessary for the proper regulation of cell cycle checkpoint by helping in the chromosomal segregation[9]. Mutations that disrupt the SKA complex can result in cell death in dividing cells[10]. Moreover, SKA2 has previously been identified as a glucocorticoid receptor (GR) interaction partner in peripheral cells in vitro, enhancing the receptor's translocation to the nucleus[11]. Single nucleotide polymorphisms and DNA methylation within the *SKA2*, as well as gene expression alterations, have been associated with posttraumatic stress disorder and suicide risk in human patients[12–18].

In the current convergent studies, we demonstrate that SA regulates neuroinflammation-mediated neurodegeneration via SKA2 and FKBP5 signaling. SKA2 inhibits SA-dependent IL-1β release by counteracting FKBP5 function in cells, mice, and human postmortem brains. Specifically, hyperactivated SA, induced by knockdown of *Ska2* contributes to an inflammatory feed-forward vicious cycle resulting in Gasdermin D (GSDMD)-mediated neurotoxicity that ultimately drives neurodegeneration.

## Results

### SKA2 acts as a molecular roadblock for secretory autophagy that inhibits vesicle-plasma membrane fusion

The final step in SA that allows for cargo secretion (e.g., IL-1β release) is the SNARE complex formation between the R-SNARE SEC22B of the secretory autophagosome and the $Q_{abc}$-SNARE complex, formed by the synaptosomal-associated proteins SNAP23 and SNAP29 and the syntaxins 3 and 4 (STX 3/4) of the plasma membrane. This event leads to the fusion of autophagosome and plasma membrane, with subsequent release of cargo proteins into the extracellular milieu[6,7,19,20].

First, we set out to investigate whether SKA2 interacts with components of the SA pathway in the brain. Co-immunoprecipitations (co-IPs) from mouse prefrontal cortex (PFC), hippocampus, and amygdala tissue, showed that SKA2 associates with SNAP29 (Fig. 1A), which has previously only been reported in HeLa cells[8]. In addition, co-IPs revealed associations of SKA2 with other SNARE complex proteins including SEC22B, SNAP23 and STX3, but not to SNARE-scaffolder FKBP5 (Fig. 1A). Protein pull-down assays, using recombinant proteins, confirmed a protein-protein interaction between SKA2 and SNAP29 (Fig. 1B). Interestingly, the assays did not reveal any direct interactions between SKA2 and either SNAP23, STX3 (or STX4) (Fig. 1B). This suggests the associations between SKA2 and these proteins might be bridged via SNAP29 given they are known to interact with each other as part of the SNARE-machinery. Moreover, additional pull-down assays confirmed a direct protein-protein interaction between FKBP5 and SEC22B (Fig. S1), which is in line with previous IP/co-IP and interactomics findings[7].

To further examine a potential role for SKA2 in the RQ-SNARE fusion process during SA, we performed co-IPs in a murine microglia cell line (SIM-A9 cells). Knockdown (KD) of *Ska2* enhanced RQ-SNARE complex formation, which was reflected by increased SEC22B binding to SNAP29 as well as SEC22B binding to STX3. Consistent with our previous findings in SH-SY5Y cells[7], overexpression (OE) of *Fkbp5* led to a similar increase in binding of SEC22B to SNAP29, as well as SEC22B to STX3 (Fig. 1C-H).

Next, we tested a functional link between SKA2 and FKBP5, and found that knockout (KO) of *Fkbp5* led to a significant increase in SKA2 to SNAP29 binding, while *Fkbp5* overexpression had the opposite effect (Fig. 1I, J). Importantly, co-IPs revealed that overexpression of *Ska2* significantly reduced FKBP5 to SEC22B binding (Fig. 1K, L), while a *Ska2* KD induced the opposite effect (Fig. 1C, M). Together, these data demonstrate that SKA2 is directly involved in RQ-SNARE complex formation and appears to regulate SA in opposition to the role of FKBP5. Interestingly, co-immunopreciptiation experiments (SNAP29-IP) in tissue homogenates from the mouse hippocampus did not reveal any associations of SKA2's partner proteins SKA1 and SKA3 of the SKA complex with SNAP29 (Fig. S2). This suggests that in SA SKA2 may act independent of SKA1 and SKA3. It is worth mentioning that the expression of SKA1 and SKA3 are relatively low in mouse brain tissue compared to SKA2.

### Activation of SA increases cargo protein release in vitro and in vivo

IL-1β is a well-established cargo protein released via SA[6]. In order to investigate whether SKA2 alters IL-1β secretion, we manipulated *Ska2* expression in (Lipopolysaccharide (LPS) and L-leucyl-L-leucine methyl ester (LLOMe-primed) SIM-A9 cells and analyzed the supernatants with enzyme-linked immunosorbent assay (ELISA). *Ska2* KD significantly increased release of IL-1β, 24 h after transfection (Fig. 2A). In addition, overexpression of *Ska2* resulted in reduced IL-1β release, while *Fkbp5* overexpression led to the opposite effect. Strikingly, *Ska2* overexpression was able to reverse the increase in IL-1β release induced by *Fkbp5* overexpression (Fig. 2B). Together these results suggest that SKA2 and FKBP5 play contrasting roles in the final step of SA, in particular with regards to IL-1β release. While FKBP5 enhances the formation of the RQ-SNARE complex and subsequent IL-1β release, SKA2 decreases it, thereby acting as gatekeeper of this secretory pathway (Fig. 2C).

Dex-induced release of the cargo protein Cathepsin D through SA is tightly linked to ATG5 function, a protein of the autophagy core machinery[7]. Previously, we have established a clear role for FKBP5 in both macroautophagy and specifically SA, as demonstrated by the increase in early autophagy markers and autophagy flux in primary astrocytes overexpressing FKBP5[21], as well as the absence of the DEX-induced Cathepsin D release in SIM-A9 *Fkbp5*-KO cells[7], respectively. These results point to the importance of functional autophagy at various steps impacting the release of SA cargo proteins. To confirm that the secretion of known SA cargo proteins is dependent on the autophagic machinery, we first assessed the levels of Cathepsin D by analyzing the supernatant of microglia cultures (SIM-A9 cells). The analysis aimed to assess Cathepsin D levels subsequent to SA induction via LLOMe treatment, along with treatment of the autophagy inhibitors SAR405 (VPS34 inhibitor, VPS34i) or MRT68921 (ULK1 inhibitor, ULK1i), respectively. At the level of secreted Cathepsin D, co-treatment of cells with 0.25 mM LLOMe and 1 μM ULK1i already abolished the significant LLOMe-induced Cathepsin D release, which was further reduced to baseline levels with 10 μM ULK1i (Fig. S3A, B). Interestingly, in cells co-treated with LLOMe and VPS34i, the inhibition of PIK3C3/Vps34 could only weakly reduce the levels of released Cathepsin D. It is possible that the additional inhibitory effect of ULK1i against AMPK and TBK1[22] further diminishes SA, pointing towards the importance of functional autophagy regulation at different levels. Although both

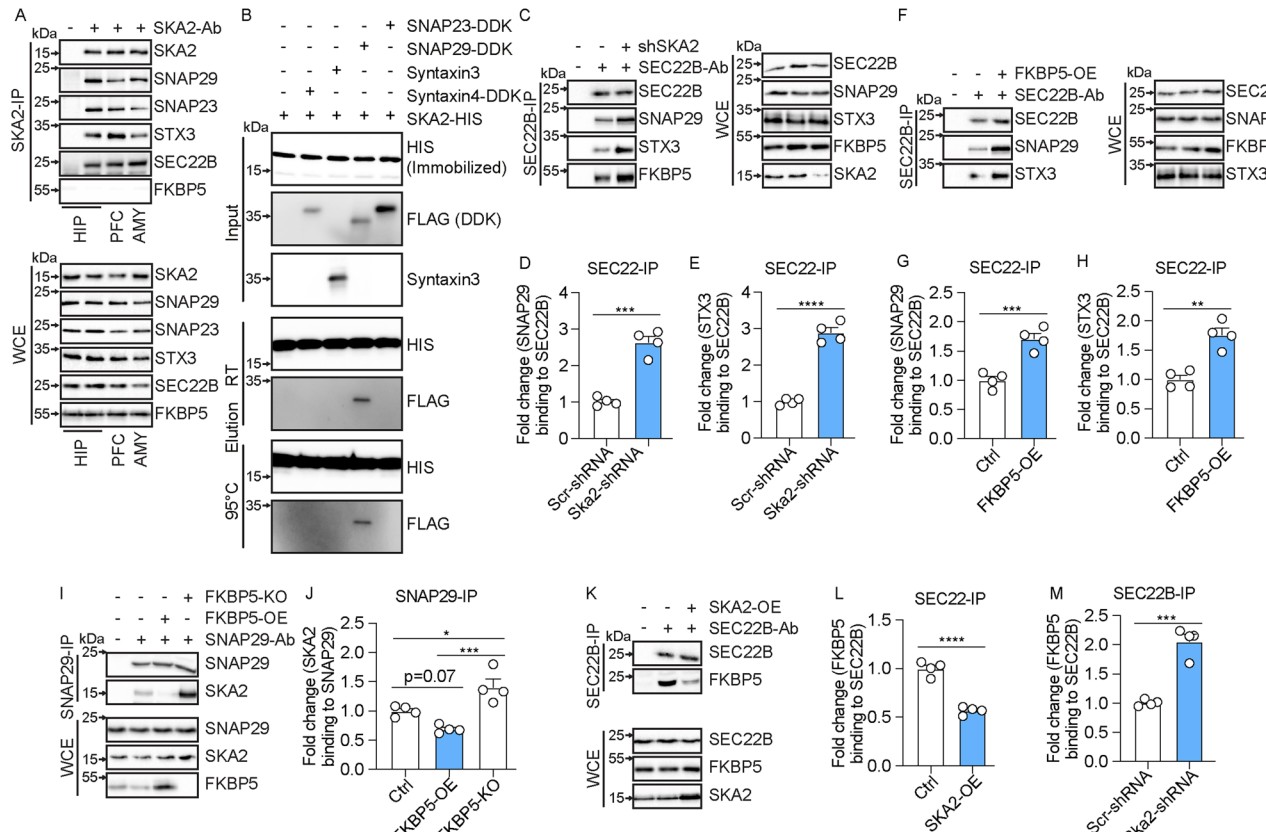

**Fig. 1 | SKA2 and FKBP5 have opposing roles in the final step of secretory autophagy (SA). A** SNAP29, SNAP23, STX3, SEC22B, and FKBP5 co-immunoprecipitation (SKA2 IP) and whole cell extract (WCE) in hippocampus (HIP), prefrontal cortex (PFC) and amygdala (AMY) samples of mice ($n = 8$). **B** HIS pull down assay (replicated in 3 independent in vitro experiments). DDK(Flag)-tagged SNAP23, SNAP29, Syntaxin3 or Syntaxin4 was incubated with purified magnetic beads-HIS-tagged SKA2 or magnetic beads-HIS protein alone. After incubation, bead bound proteins were eluted at room temperature (RT) or at 95 °C and subjected to western blot analysis using antibodies against HIS and FLAG. Input lane contains HIS alone (left) or HIS-tagged SKA2 (right). **C−M** SIM-A9 cells transfected with SKA2, FKBP5 or their respective controls, were harvested 24 h later. After immunoprecipitation (IP) of protein complexes, input and co-IP

proteins were quantified by western blotting. **C, F, I, K** Representative blots of (**D, E, G, H, J, L, M**). Graphs display quantification of SNAP29/SEC22B, STX3/SEC22B, SKA2/SNAP29, FKBP5/SEC22B protein association after SEC22B or SNAP29 IP (unpaired two tailed t-test: (**D**) $t_6 = 8.945$, $p < 0.0001$, (**E**) $t_6 = 12.94$, $p < 0.0001$, (**G**) $t_6 = 6.056$, $p = 0.0009$, (**H**) $t_6 = 5.554$, $p = 0.0014$; one-way ANOVA: (**J**) $F_{2, 9} = 17.28$, $p = 0.0008$, Tukey's post hoc test: ctrl vs. FKBP5-OE, $p = 0.0743$, ctrl vs. FKBP5-KO, $p = 0.0218$, FKBP5-OE vs. FKBP5-KO, $p = 0.0006$; unpaired two tailed t-test: (**L**) $t_6 = 10.27$, $p < 0.0001$, (**M**) $t_6 = 8.140$, $p = 0.0002$; $n$ = mean derived from four independent in vitro experiments). $* = p < 0.05$; $** = p < 0.01$; $*** = p < 0.001$; $**** = p < 0.0001$. Data are presented as mean + SEM. Source data are provided as a Source Data file.

inhibitors influence the formation of autophagosomes, they might affect distinct subsets of proteins, leading to varied effects on SA. Furthermore, compensatory mechanisms have to be considered, as one inhibitor might trigger compensatory responses that affect SA differently than compensatory responses triggered by the other inhibitor. To demonstrate the inhibitory activity of ULK1i and VPS34i against autophagy, we performed autophagy flux measurements in the absence and presence of Bafilomycin A1 (BafA1), aiming for LC3B-lipidation in SIM-A9 cell cultures. Both compounds led to a dose-dependent reduction of autophagy flux, evidenced by a decrease in LC3B-II protein levels, with VPS34i being more potent even at lower doses (0.1 and 1 μM) compared to ULK1i (Fig. S3C, D). Together these findings, based on pharmacological and genetic manipulations, high-light the extensive signaling crosstalk of autophagy pathways that impact SA and demonstrate the effectiveness of Atg5-KO[7], Fkbp5-KO[7], and ULK1 inhibition in influencing SA.

To confirm that secretion of known SA cargo proteins is dependent on the autophagic machinery in vivo, we assessed IL-1β's extracellular dynamics in medial PFC (mPFC) using in vivo microdialysis in C57Bl/6NCrl mice injected with the early autophagy inhibitor ULK1i. Microdialysates were collected under baseline conditions and following acute and strong foot shock stress with the aim to potentiate IL-1β

release (Fig. 2D, E). We previously showed that acute stress increases activity of SA and subsequent release of Cathepsin D[7] in vivo. There were no changes in IL-1β secretion under baseline conditions between the treatment groups. In contrast, stress-induced IL-1β release was significantly decreased in mice treated with ULK1i compared to vehicle controls (Fig. 2F). Along these lines, acute stress induced a significant increase in IL-1β secretion in wild type mice, an effect that was blunted in *Fkbp5* KO mice (Fig. 2G). These data further validate our in vitro findings of IL-1β secretory regulation. They also underline the importance of the SA pathway on brain physiology and the potential impact on neuroinflammation.

## Hyperactivity of SA leads to NLRP3 inflammasome activation, neuroinflammation-induced neurodegeneration

In order to better understand the relevance of SA and its impact on brain physiology, we performed a viral-mediated shRNA-dependent KD of *Ska2* in the hippocampus of C57Bl/6J mice (Fig. S4A). Remarkably, KD of *Ska2* induced pronounced neurodegeneration compared to viral infection with a scrambled control shRNA (Fig. 3). *Ska2* KD resulted in complete hippocampal atrophy within 6 weeks of the viral injection (Fig. 3A). This was also reflected in decreased expression of the neuronal marker NeuN and drastically reduced CA1 thickness,

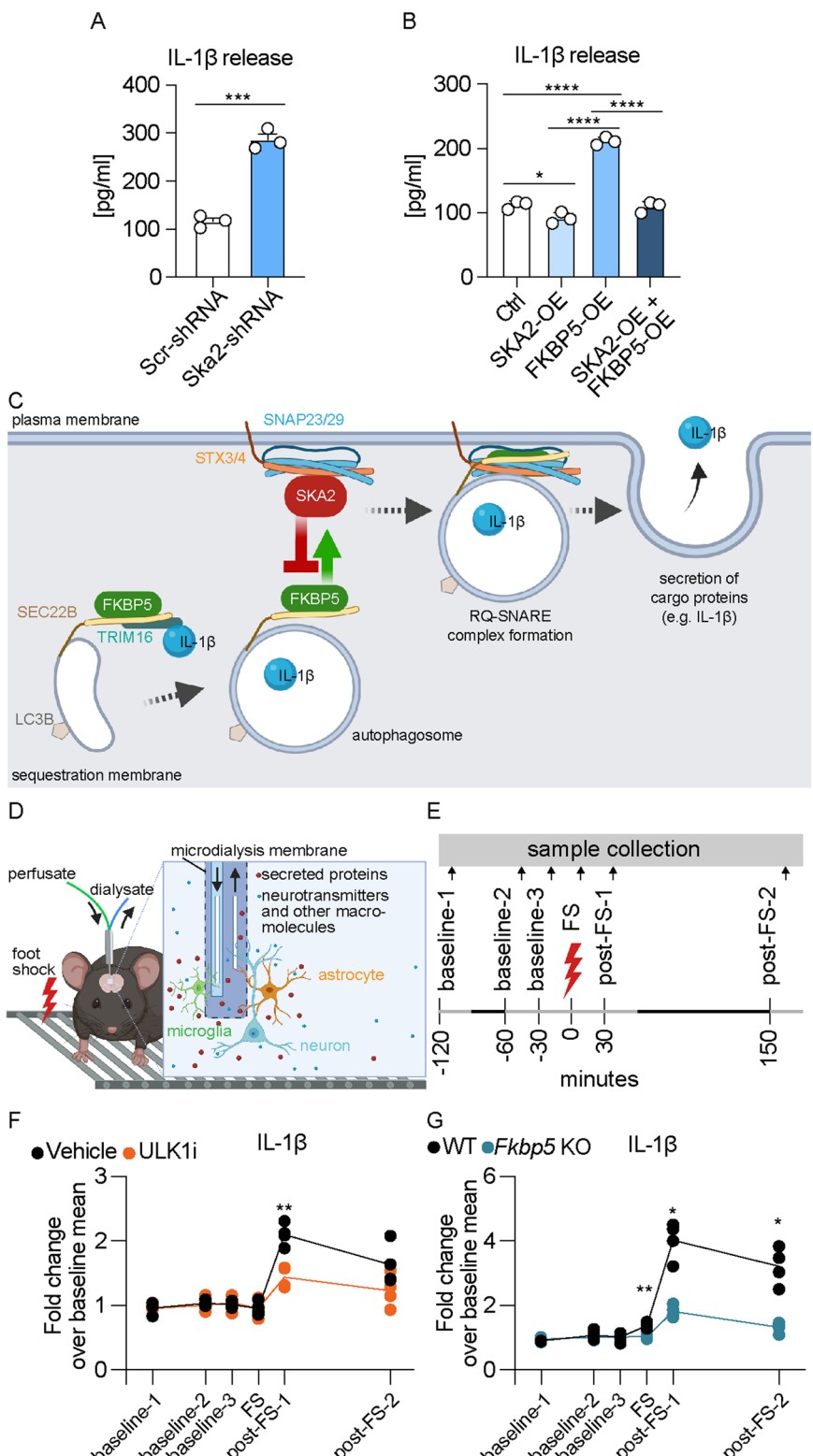

two- and 4-weeks following KD of *Ska2* (Fig. 3B). In addition, immunohistochemistry (IHC) with IBA1 revealed an increase in microglia numbers in the hippocampus, 2 weeks after viral-mediated KD of *Ska2*, an effect that was even more pronounced after 4 weeks (Fig. 3C). Moreover, expression of the astrocyte marker GFAP was increased 2 and 4 weeks following *Ska2* KD in the hippocampus (Fig. 3D). No assessment of marker expression was possible at week 6 due to complete hippocampal atrophy. Importantly, the neurodegenerative process and inflammatory response were not caused by off-target effects

since a similar phenotype was observed upon KD of *Ska2* with a second *Ska2*-shRNA, targeting a different region of the gene and packaged into a different viral capsid serotype (Fig. S4B, C). Taken together, these findings indicate that hippocampal disruption of SKA2 leads to progressive neuroinflammation and subsequent neurodegeneration, likely through an overactivation of the SA pathway.

Previously, increasing intensities of pro-inflammatory stimuli (e.g., microbial components or endogenous cytokines) have been shown to induce sequential activation of vesicular and GSDMD-mediated IL-1β

**Fig. 2 | Activation of secretory autophagy (SA) increases cargo protein release in vitro and in vivo. A, B** IL-1β release measured via ELISA from supernatants of SIM-A9 cells 24 h after manipulation of SKA2 and/or FKBP5 expression, and following overnight LPS (100 ng/mL) and treatment with LLOMe (0.25 mM) for 3 h (unpaired two tailed t-test: (A) $t_4 = 11.99$, $p = 0.0003$; one-way ANOVA: **B** $F_{3, 8} = 158.6$, $p < 0.0001$; Tukey's post hoc test: ctrl vs. SKA2-OE, $p = 0.0384$, ctrl vs. FKBP5-OE, $p < 0.0001$, SKA2-OE vs. FKBP5-OE, $p < 0.0001$, FKBP5-OE vs. SKA2 + FKBP5 OE, $p < 0.0001$; $n =$ mean derived from three independent in vitro experiments).
**C** Schematic overview of the SA pathway with SKA2 and FKBP5. The cargo receptor TRIM16, together with SEC22B, transfers molecular cargo (e.g., IL-1β) to the autophagy-related LC3B-positive membrane carriers. SEC22B, now acting as an R-SNARE on the delimiting membrane facing the cytosol, carries out fusion at the plasma membrane in conjunction with the Q$_{bc}$-SNAREs, SNAP23 and SNAP29 (SNAP23/29), and one of the plasma membrane Q$_a$-SNAREs, STX3 or STX4 (STX3/4), thus delivering IL-1β to the extracellular milieu, where it exerts its biological functions. FKBP5 acts as a positive regulator of SA by enhancing TRIM16-SEC22B

complex formation as well as autophagosome-plasma membrane fusion via the SNARE-protein complex assembly. In contrast, SKA2 inhibits the SNARE-protein complex formation during vesicle-plasma membrane fusion, thereby acting as gatekeeper of SA. **D, E** Schematic overview of in vivo microdialysis and the experimental design and timeline; each sample was collected over 30 min indicated by the light gray lines. Quantifications of IL-1β, determined by capillary-based immunoblotting from in vivo medioprefrontal cortex microdialysis of C57Bl/6NCrl mice injected intraperitoneally with ULK1 inhibitor (ULK1i, an autophagy inhibitor) or saline (**F**; repeated measures two-way ANOVA, time × treatment interaction: $F_{5, 30} = 7.064$, $p = 0.0002$; Šidák's multiple comparisons post hoc test, post-FS-1: $p = 0.0084$; $n = 4$ mice per group) as well as of wild type (WT) and global *Fkbp5* knockout mice (**G**; repeated measures two-way ANOVA, time × genotype interaction: $F_{5, 30} = 34.15$, $p < 0.0001$; Šidák's multiple comparisons post hoc test: FS: $p = 0.009$, post-FS-1: $p = 0.0163$, post-FS-2: $p = 0.0294$; $n = 4$ mice per group). FS foot shock. * = $p < 0.05$; ** = $p < 0.01$; *** = $p < 0.001$; **** = $p < 0.0001$. Data are presented as mean + SEM. Source data are provided as a Source Data file.

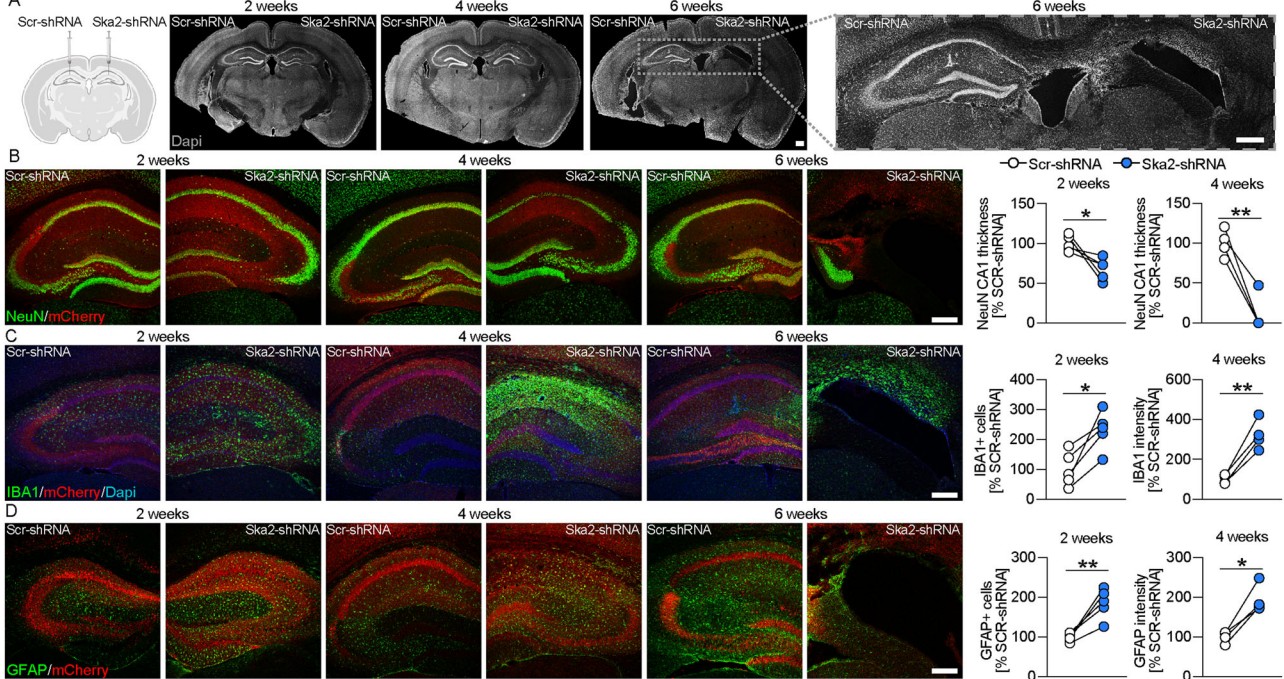

**Fig. 3 | Hippocampal *Ska2* knockdown induces neuroinflammation-mediated neurodegeneration in mice. A** Schematic representation of viral injections (Scr-shRNA-AAV and Ska2-shRNA-1-AAV) into the hippocampus (left). (right) Representative IHC images of DAPI (gray) 2, 4 and 6 weeks after viral injections. **B** IHC images of NeuN (green) and mCherry (red, viral marker) 2, 4, and 6 weeks after viral injection. Quantification of CA1 thickness 2 and 4 weeks after viral injection (paired t-test: 2 weeks, $t_4 = 3.194$, $p = 0.0331$; $n = 5$ mice; 4 weeks, $t_3 = 6.711$, $p = 0.0068$; $n = 4$ mice). **C** IHC images of IBA1 (green), mCherry (red), and DAPI (blue) 2, 4, and

6 weeks after viral injection. Quantification of IBA1 expression 2 and 4 weeks after viral injection (paired t-test: 2 weeks, $t_4 = 4.295$, $p = 0.0127$; $n = 5$ mice; 4 weeks, $t_3 = 7.165$, $p = 0.0056$; $n = 4$ mice). **D** IHC images of GFAP (green) and mCherry (red) 2, 4, and 6 weeks after viral injection. Quantification of GFAP expression 2 and 4 weeks after viral injection (paired t-test: 2 weeks, $t_4 = 5.524$, $p = 0.0052$; $n = 5$; 4 weeks, $t_3 = 5.764$, $p = 0.0104$; $n = 4$ mice). * = $p < 0.05$; ** = $p < 0.01$; Scale bars represent 250 μm. Source data are provided as a Source Data file.

secretory pathways[23]. In order to investigate whether altered SA activity (and thus IL-1β release) is able to modulate inflammasome formation, we used a clonal inflammasome reporter overexpressing fluorescently tagged ASC (apoptosis-associated speck-like protein containing a CARD)[24] in wild type and *Sec2b* KO SIM-A9 cells. Already under control conditions, inhibition of SA activity (through *Sec2b* KO) resulted in significantly less ASC specks compared to wild type controls (Fig. 4A). Moreover, ASC specks were significantly increased in SIM-A9 WT cells following LPS treatment or KD of *Ska2* compared to vehicle or Scr-shRNA controls (Fig. 4B). In contrast, the LPS- and *Ska2* KD-dependent inflammasome formation was abolished when the SA pathway was disrupted in *Sec2b* KO SIM-A9 cells (Fig. 4C). In order to identify which inflammasome is stimulated through increased activity of SA, we investigated protein lysates of organotypic hippocampal slice cultures.

KD of *Ska2* resulted in significantly increased binding of SEC22B to SNAP29, reflective of enhanced SA activity (Fig. 4D). The kinase NEK7 is an important requirement in the activation of the NLRP3 (NOD-, LRR- and pyrin domain-containing protein 3) inflammasome via NLRP3-NEK7 association[25]. Along these lines, NEK7 binding to NLRP3 was significantly increased following *Ska2* KD (Fig. 4E). Therefore, we next investigated whether KD of *Ska2* in the hippocampus of mice and thus overactivation of SA, may serve as an inflammasome-inducing signal leading to GSDMD-mediated IL-1β secretion. Indeed, KD of *Ska2* led to increased ASC expression and ASC specks formation (Fig. 4F, G) as well as CASPASE-1 (CASP-1) expression (Fig. 4H, I), indicative of inflammasome activation. Inflammasome-activated CASP-1 cleaves GSDMD to release the N-terminal domain which forms pores on the membrane that enable passage of cytokines including IL-1β[26,27]. Accordingly, the expression

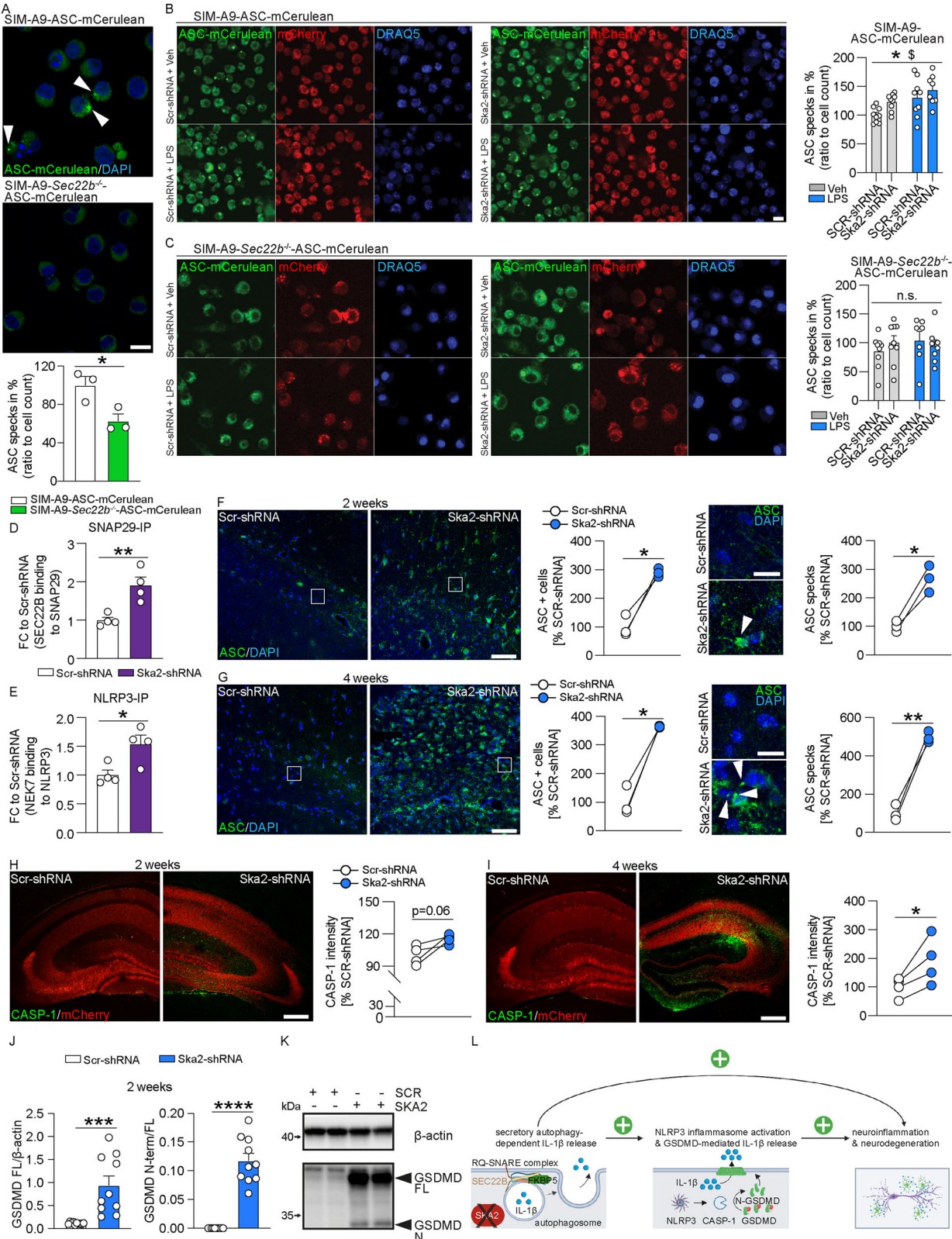

levels of full length (FL) GSDMD as well as its cleaved N-terminal domain (GSDMD N-term) were increased at 2 weeks following KD of *Ska2* (Fig. 4J, K). Together, these data provide significant mechanistic evidence that hyperactivated SA (through KD of *Ska2*) is able to create an inflammatory feed-forward vicious cycle resulting in a GSDMD-mediated excessively neurotoxic environment to ultimately catalyze neuroinflammation and neurodegeneration (Fig. 4L).

## *Ska2* knockdown in the hippocampus leads to cognitive impairment

The severe hippocampal atrophy observed at 4 weeks following *Ska2* KD resulted in expected spatial memory (Y-maze) and novel object recognition memory impairments in mice (Fig. 5A, B). The observed cognitive deficits were not accompanied by changes in general locomotor activity (open field test) or anxiety-related behavior (elevated

**Fig. 4 | Hyperactivity of SA induces inflammasome formation. A** SIM-A9 *Sec22b⁻/⁻* cells expressing ASC (apoptosis-associated speck-like protein containing a CARD) -mCerulean (via epifluorescence) show a significantly decreased number of intracellular (white arrows) ASC specks compared to wild type (WT) SIM-A9 cells (unpaired two tailed t-test: $t_4 = 3.206$, p = 0.0327; $n$ = mean derived from three independent in vitro experiments). **B** In WT SIM-A9 cells knockdown of *Ska2* or LPS treatment leads to a significantly increased number of intracellular ASC specks compared to Scr-shRNA or LPS-treated cells (2-way ANOVA: main LPS treatment effect ($), $F_{1,31} = 10.60$, $p = 0.0027$, main *Ska2* knockdown effect (*), $F_{1,31} = 5.482$, $p = 0.0258$; $n$ = 9 WT Veh SCR-shRNA, $n$ = 9 WT Veh SKA2-shRNA, $n$ = 9 WT LPS SCR-shRNA, $n$ = 8 WT LPS SKA2-shRNA). **C** In contrast, knockdown of *Ska2* or LPS treatment does not have any effects on the number of ASC specks in SIM-A9 *Sec22b⁻/⁻* cells (2-way ANOVA: n. s. treatment effect $F_{1,29} = 0.312$, $p = 0.5804$, main *Ska2* knockdown effect, $F_{1,29} = 0.055$, $p = 0.8157$; $n$ = 9 for SEC22B KO Veh SCR-shRNA and SKA2-shRNA, $n$ = 7 SEC22B KO LPS SCR-shRNA, $n$ = 8 SEC22B KO LPS SKA2-shRNA). **D, E** Knockdown of *Ska2* leads to significantly increased SEC22B binding to SNAP29 (unpaired two tailed t-test: $t_4 = 4.113$, $p = 0.0063$; $n$ = 4 independent biological replicates) as well as NEK7 binding to NLRP3 in protein lysates of organotypic hippocampal slice cultures (unpaired two tailed t-test: $t_4 = 2.998$, $p = 0.0241$; $n$ = 4 independent biological replicates). **F** IHC images of ASC (green) and DAPI (blue) 2 weeks after viral injection (Scr-shRNA-AAV and Ska2-shRNA-1-AAV) into the hippocampus. Quantification of ASC+ cells (left) and ASC specks (right) 2 weeks after viral injection (paired t-test: ASC+ cells, $t_2 = 6.414$, p = 0.0235, ASC specks, $t_2 = 6.937$, p = 0.0202; n = 3 mice). **G** IHC images of ASC (green) and DAPI (blue) 4 weeks after viral injection (Scr-shRNA-AAV and Ska2-shRNA-1-AAV) into the hippocampus. Quantification of ASC+ cells (left) and ASC specks (right) 4 weeks after viral injection (paired t-test: ASC+ cells, $t_2 = 8.511$, $p = 0.0135$; ASC specks, $t_2 = 10.99$, $p = 0.0082$; $n$ = 3 mice). **H** IHC images of CASPASE-1 (CASP-1) (green) and mCherry (red, viral marker) 2 weeks after viral injection (Scr-shRNA-AAV and Ska2-shRNA-1-AAV) into the hippocampus (left). (right) Quantification of CASP-1 expression 2 weeks after viral injection (paired t-test: $t_3 = 2.842$, $p = 0.0655$, $n$ = 4 mice). **I** IHC images of CASP-1 (green) and mCherry (red, viral marker) 4 weeks after viral injection (Scr-shRNA-AAV and Ska2-shRNA-1-AAV) into the hippocampus (left). (right) Quantification of CASP-1 expression 4 weeks after viral injection (paired t-test: $t_3 = 3.367$, $p = 0.0435$, $n$ = 4 mice). **J** Full length Gasdermin D (GSDMD FL) levels as well as the ratio of the cleaved N-terminal form of GSDMD (GSDMD N-term) to GSDMD FL are increased 2 weeks after *Ska2* knockdown (unpaired two tailed t-test; GSDMD FL/ β-actin: $t_{18} = 4.105$, $p = 0.0007$, GSDMD N-term/GSDMD FL: $t_{18} = 9.259$, $p < 0.0001$; $n$ = 10 independent biological replicates per group). **K** Examples blots of (**E**). **L** Schematic overview of the interaction between secretory autophagy (SA) and the GSDMD-mediated IL-1β release. SKA2 depletion results in increased SA-dependent IL-1β release, serving as a molecular vicious feed-forward loop for inflammasome activation. Inflammasome assembly activates CASP-1 enzymatic function. ASC in the inflammasome complex recruits CASP-1. Activation of CASP-1 cleaves GSDMD to release the N-terminal domain, which forms pores in the plasma membrane for uncontrolled IL-1β release. * = $p$ < 0.05; ** = $p$ < 0.01; *** = $p$ < 0.001, **** = $p$ < 0.0001. Data are presented as mean + SEM. Scale bar represents 5 μm in A, 50 μm in (**F, G**) (left), 10 μm in (**B, F, G**) (right), and 250 μm in (**H, I**). Source data are provided as a Source Data file.

plus maze (EPM)), which confound learning and memory tasks (Fig. 5C, D). These findings indicate that hippocampal disruption of SKA2 leads to cognitive impairment.

## RNA sequencing analyses following hippocampal knockdown of S*ka2* identify transcriptional signatures associated with increased activity of secretory autophagy and cell death processes

To further address the causal mechanisms of SKA2-mediated neurodegeneration, we employed RNA sequencing of hippocampal tissue 2 and 4 weeks post viral-mediated KD of *Ska2*. We investigated whether transcriptional changes might be associated with a hyperactivated SA pathway and subsequent inflammasome activation, as well as with programmed cell death such as apoptosis, pyroptosis and necroptosis.

Given the neuronal and microglial IHC results of Fig. 3B, C indicated that KD of *Ska2* has an impact on the cellular composition in the hippocampus, we first deconvoluted the bulk RNA-seq data with the multi-subject single-cell (MuSiC) method[28]. Confirming the IHC results (Fig. 3B, C), *Ska2* KD led to altered estimated cell proportions in the hippocampus including decreased numbers of neurons and increased numbers of microglia at 2 weeks. These changes were even more pronounced at the 4-week time point (Fig. 6A, B). In order to investigate the transcriptional changes associated with *Ska2* KD, and to distinguish these from potential transcriptional changes due to alterations in cell type proportions, we corrected the bulk-RNA-seq data based on the changes in the proportion of cell types. Given that the changes in estimated cell proportions were highly correlated across the different cell types (Fig. S5), we corrected the change in cell type composition using the estimated microglia proportion which is representative of the change of all other cell types.

At the 2-week time point, differential expression analysis identified 1367 differentially expressed genes (DEGs) ($p$ < 0.05 adjusted with Bonferroni correction) of which 267 genes were upregulated and 18 downregulated with a Log₂ Fold Change (FC) > 2 (Fig. 6C, left, Supplementary Data 1). We found that many of the DEGs are associated with an immune response. Interestingly, numerous upregulated genes belong to classes of immune mediators such as chemokines, cathepsins and cytokines. Notably, several genes (Fig. 6C, right) including *Il-1β, Il-18, Ccl4, Ccl5, Ccl9, and Ctsz* have previously been shown to be released through SA[6,7]. Thus, increased release of cargo proteins

through SA from intracellular pools induced by KD of *Ska2* might lead to compensation through increased mRNA expression and thus production of the secreted proteins. Interestingly, at the 4-week time point, differential expression analysis identified only 601 DEGs ($p_{adj}$ < 0.05) after *Ska2* KD, of which 46 genes were upregulated and 68 downregulated with a Log₂FC > 2 (Fig. 6D, Supplementary Data 2) when corrected for estimated changes in the cell type composition.

Next, we performed a Gene Ontology (GO) enrichment analysis which revealed numerous enriched terms of biological processes, cellular components and molecular functions at 2 weeks following *Ska2* KD. For the entire list of enriched GO terms please refer to Supplementary Data 3. Many of these GO terms such as "cytokine-mediated signaling pathway", "regulation of cytokine-mediated signaling pathway", "positive regulation of interleukin-1 beta production" and "macrophage activation" from the 2-week time point can be associated with increased activity of SA before substantial neuronal degeneration and cell death (Fig. 6E). In addition, the GO terms "regulation of inflammatory response" and "positive regulation of innate immune response" were significantly enriched further supporting that cells are poised for inflammasome activation and subsequent increase in uncontrolled IL-1β secretion via the GSDMD pathway (i.e., pyroptosis). Moreover, the enriched GO term "lysosome" is in support of changes in SA following *Ska2* KD. Notably, multiple of the immune-related enriched GO terms such as "positive regulation of tumor necrosis factor superfamily cytokine production", and "phagocytosis" also point towards an increased apoptotic activity, which is consistent with a previous report that mutations disrupting the SKA complex can result in cell death[10]. Interestingly, at the 4-week time point the GO enrichment analysis with the DEGs revealed only the following three enriched terms of biological processes and cellular components "glial cell differentiation", "external encapsulating structure" and "extracellular matrix" (Fig. S6, Supplementary Data 4).

Next, we conducted a Kyoto Encyclopedia of Genes and Genomes (KEGG) pathway enrichment analysis with the DEGs of the 2-week time point. The results showed that the DEGs were highly associated with several immune pathways including "cytokine-cytokine receptor interaction", "NOD-like receptor signaling pathway", "Toll-like receptor signaling pathway" and "Chemokine signaling pathway", (Fig. 6F, Supplementary Data 5) which can play a role on neurodegenerative processes. In contrast, KEGG pathway

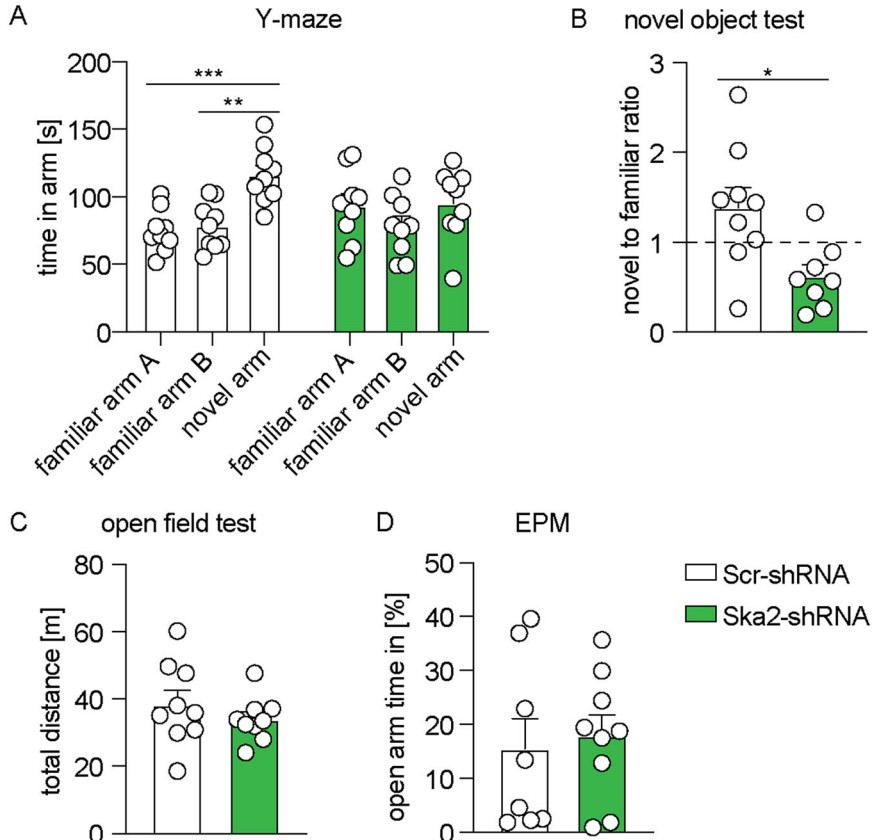

**Fig. 5 | Hippocampal *Ska2* knockdown leads to cognitive impairment in mice.** In the Y-maze test, mice injected with Scr-shRNA spent significantly more time in the novel arm compared to both familiar arms (**A**, **B**). These effects were abolished following *Ska2* knockdown (2-way ANOVA: condition × arm interaction, $F_{2,48} = 3.626$, $p = 0.0342$, Tukey's post hoc test: familiar arm A vs. novel arm, $p = 0.0007$, familiar arm B vs. novel arm, $p < 0.002$; $n = 9$ per group). **B** In contrast to control animals, Ska2-shRNA mice did not discriminate between a novel and familiar object during the novel object recognition test (unpaired t-test: $t_{15} = 2.840$, $p = 0.0124$; $n = 9$ Scr-shRNA mice, $n = 8$ Ska2-shRNA mice). **C, D** *Ska2* knockdown did not alter general locomotor activity in the open field test (unpaired t-test: $p > 0.05$, $n = 9$ per group) or anxiety-related behavior in the elevated plus maze (EPM) (unpaired t-test: $p > 0.05$, $n = 8$ Scr-shRNA mice, $n = 9$ Ska2-shRNA mice). $* = p < 0.05$; $** = p < 0.01$; $*** = p < 0.001$. Data are presented as mean + SEM. Source data are provided as a Source Data file.

enrichment analysis of the DEGs did not reveal any enriched terms at 4 weeks.

Collectively, the RNA-seq analyses provide compelling additional evidence that SKA2 acts a critical negative regulator of the SA pathway in the brain. These findings support the hypothesis that hyperactivation of the SA pathway – triggered by *Ska2* KD, especially notable at the 2-week time point—initiates an inflammatory feed-forward loop. This loop leads to an excessively neurotoxic environment, which not only catalyzes neuroinflammation and neurodegeneration but also dramatically alters the cellular landscape of the hippocamps. Furthermore, it unveils the intricate involvement of SKA2 in various functions spanning multiple pathways, which collectively contribute to neuroinflammation and the process of neurodegeneration and cell death (apoptosis, necroptosis and pyroptosis), both through direct and indirect mechanisms.

**Secretory autophagy is increased in human postmortem Alzheimer's disease samples**

Given the impact that SA and its regulators, SKA2 and FKBP5, have on brain function in mice, we continued to explore the relevance of this secretory pathway and its components in the human brain. To investigate the relationship of the *SKA2* and *FKBP5* genes with phenotypic traits, we searched these loci in the Atlas of genome-wide association studies (GWAS) Summary Statistics (http://atlas.ctglab.nl/PheWAS)[29]. Interestingly, Phenome-Wide Association Studies (PheWAS) associated the *FKBP5* locus with, among others, immunological traits such

as lymphocyte count, white blood cell count and monocyte percentage of white cells. PheWAS of the *SKA2* locus associated with cognitive as well as with immunological traits, including intelligence and cognitive performance as well as monocyte percentage of white cells, granulocyte percentage of myeloid white cells and monocyte count (Fig. S7A, B, Supplementary Data 6, 7).

Next, performing co-IPs, we confirmed an association of SKA2 with SNAP29 in human PFC, amygdala, and hippocampus in postmortem tissue from healthy subjects (Fig. 7A, Supplementary Data 8). IHC of brain sections from healthy human subjects revealed a pronounced expression of SKA2 in the adult hippocampus (mid-body coronal sections consisting of the dentate gyrus and the stratum oriens and pyramidal cell layers of the CA1, CA2, CA3, and CA4 subregions). Additional morphological and co-expression analyses revealed a prominent expression of SKA2 not only in pyramidal neurons, but also in microglia (Fig. 7B–D, Supplementary Data 9). Moreover, SKA2 co-localizes with SNAP29 on the cell surface and within the cytoplasm of neurons and microglia (Fig. 7E–I).

Given that our data suggest a critical role for SKA2 in SA and neuroinflammation-induced neurodegeneration, we further investigated whether a hyperactivated SA pathway is involved in the pathophysiology of AD. Therefore, we analyzed SKA2 protein expression using Western blotting, and performed co-IPs and subsequent capillary-based immune analysis to explore SEC22B to SNAP29 binding in the hippocampus of a cohort of AD cases ($n = 7$) and age matched controls ($n = 13$) (Supplementary Data 10). SKA2 expression was

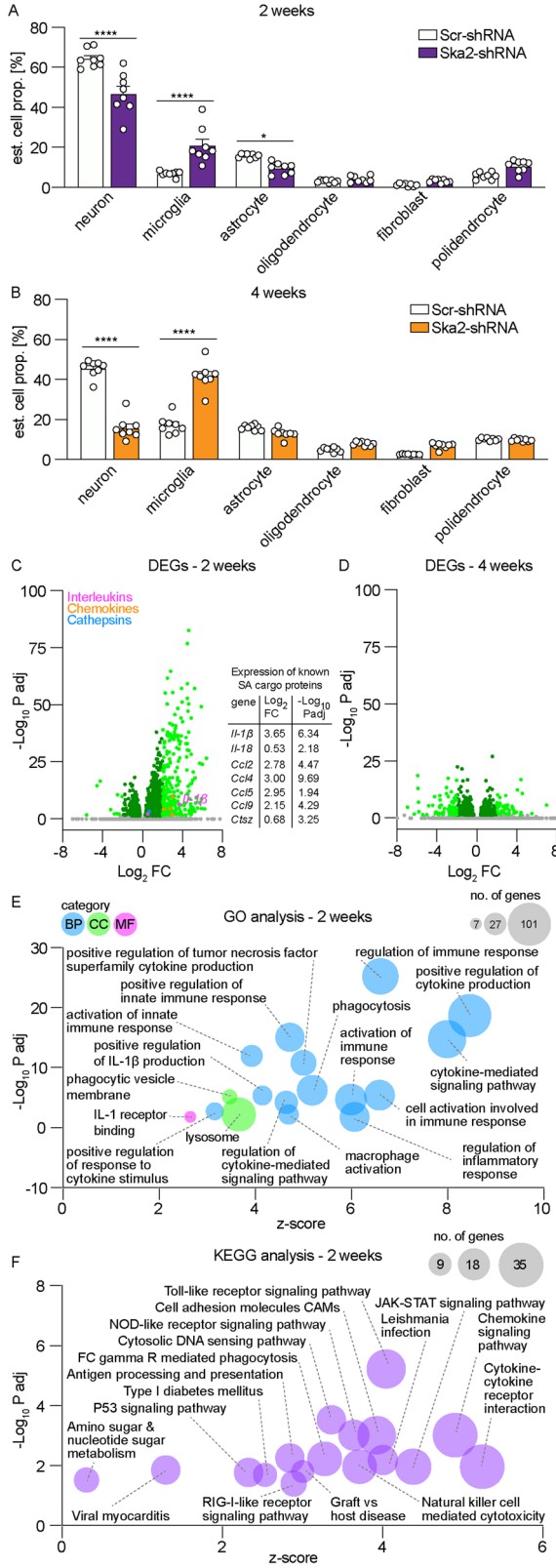

**Fig. 6 | RNA sequencing analyses following hippocampal knockdown of *Ska2* identify transcriptional signatures associated with increased activity of secretory autophagy (SA) and cell death processes.** Deconvolution analysis of the RNA sequencing data ($n = 8$ per group) at 2 weeks (**A**) and 4 weeks (**B**) following hippocampal *Ska2* knockdown reveals altered estimated cell proportions in the hippocampus (main cell types shown) (2-way ANOVA: 2 weeks: condition × cell type interaction $F_{5,84} = 23.44$, $p < 0.0001$, Bonferroni's post hoc test: neuron, $p < 0.0001$, microglia, $p < 0.0001$, astrocyte, $p = 0.0320$, $n = 8$ per group; 4 weeks: condition × cell type interaction $F_{5,84} = 115.2$, $p < 0.0001$, Bonferroni's post hoc test: neuron, $p < 0.0001$, microglia, $p < 0.0001$, $n = 8$ per group). * = $p < 0.05$; **** = $p < 0.0001$; Data are presented as mean + SEM. Differential gene expression analysis 2 (**C**) and 4 weeks (**D**) following hippocampal *Ska2* knockdown after correction for changes in the proportion in cell types. Differentially expressed genes (DEGs) ($p < 0.05$ adjusted with Bonferroni correction) are depicted in dark green (absolute Log₂FC < 2) or light green (absolute Log₂FC > 2). Magenta data points depict significant differentially expressed interleukins, while orange data points show chemokines and blue data points illustrate cathepsins. Gray data points are not significant (39 data points at 2 weeks and 25 data points at 4 weeks are outside the axis limits; for full list of all DEGs see Supplementary Data 1, 2). Gene expression changes of some known SA cargo proteins at 2 weeks are depicted in the table. **E** Gene ontology (GO) enrichment analysis with DEGs at 2 weeks. Selected enriched GO terms relevant to SA are depicted (for full list of all GO terms see Supplementary Data 3). Circle size is proportional to the number of genes. BP biological process, CC cellular component, MF molecular function. **F** Kyoto Encyclopedia of Genes and Genomes (KEGG) pathway enrichment analysis with DEGs at 2 weeks. Circle size is proportional to the number of genes. Source data are provided as a Source Data file.

cases with the lowest SKA2 expression ($n = 12$) compared AD cases with the highest SKA2 expression ($n = 12$) (Fig. 7K, middle). Moreover, NEK7 binding to NLRP3 was significantly increased in the low SKA2 expression compared to the high SKA2 expression AD group (Fig. 7K, right), indicative of augmented NLRP3 inflammasome activation.

Collectively, our data suggest an important role of SA and its regulators, FKBP5 and SKA2, in microglia and brain function. Importantly, we provide evidence for involvement of SA in inflammasome activation, neuroinflammation and the pathophysiology of AD.

## Discussion

There is a growing body of evidence that SA may be implicated in processes ranging from cancer to cell death and degeneration, due to its diverse cargo ranging from granule content to cytokines[30–34]. Moreover, a decrease in lysosomal integrity, which is a hallmark of SA[6,7,35], might subsequently reduce the function of homeostatic neuroprotective lytic autophagy. Along these lines, our results support a model in which SA differentially regulates neuroinflammation-induced neurodegeneration via SKA2 and FKBP5 signaling and is implicated in AD.

Overactivation of this pathway in mice through viral-mediated KD of hippocampal *Ska2* resulted in strong microglial activation and recruitment, leading to complete hippocampal atrophy within 6 weeks of viral injection. IL-1β is an essential cytokine, but its release needs to be strictly controlled to avoid severe inflammatory manifestations. Several pathways have been proposed to mediate its release involving secretory lysosomes, exosomes, micro-vesicles and autophagic vesicles as well as GSDMD-dependent routes[5,23,36]. Further, it has been suggested that pathways that involve the translocation of IL-1β into intracellular vesicles of lysosomal origin (that eventually fuse with the plasma membrane) are primarily in control of IL-1β release upon low pro-inflammatory stimuli, whereas stronger stimulation or concomitant cell stress induces uncontrolled secretion of IL-1β via the GSDMD-mediated pathway[23,35].

Our data suggest that hyperactivated SA, particularly in its most severe form, represents a strong enough stimulus to result in a vicious molecular feed-forward loop. This triggers the production and uncontrolled secretion of pro-inflammatory cytokines through

significantly decreased in AD, while SEC22B to SNAP29 binding was increased (Fig. 7J). Importantly, we were able to validate these findings in an independent replication cohort of PFC samples of AD cases ($n = 40$) and age matched controls ($n = 37$) (Supplementary Data 11), demonstrating significantly reduced SKA2 expression in AD (Fig. 7K, left), and thus pointing towards hyperactivated SA in AD. Along these lines, SEC22B binding to SNAP29 was significantly increased in AD

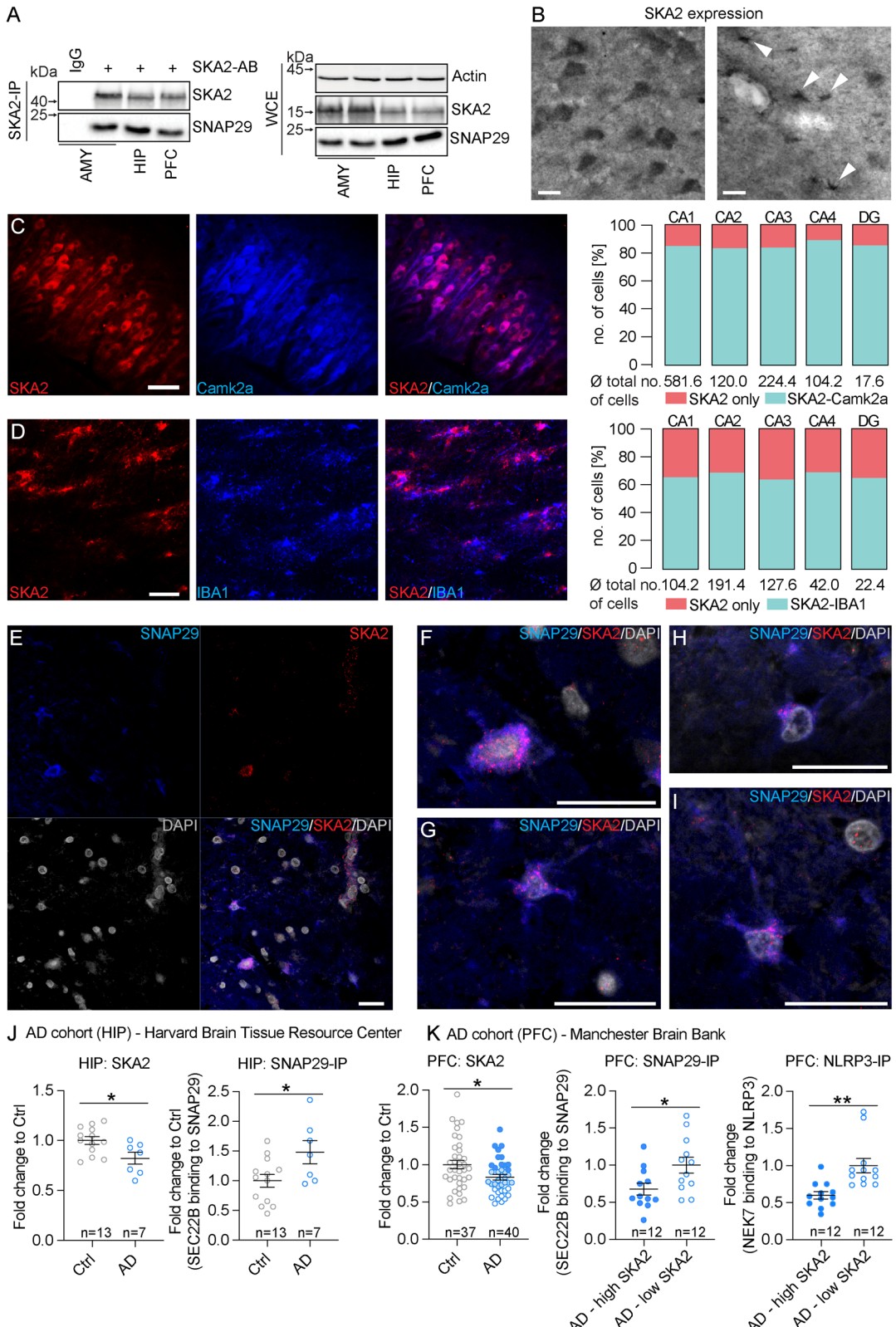

**J** AD cohort (HIP) - Harvard Brain Tissue Resource Center

**K** AD cohort (PFC) - Manchester Brain Bank

GSDMD-mediated pathways, ultimately leading to pyroptosis and neurodegeneration. Interestingly, *SKA2* DNA methylation has been linked to a decrement in thickness of the PFC[18] and less SKA2 expression in surrounding tissue[12]. Notably, our findings emphasize the presence of multiple pathways and mechanisms, encompassing various functions of SKA2[9-11], which may contribute, either directly or indirectly, to the observed effects on neuroinflammation and various

cell death processes including apoptosis, necroptosis, and pyroptosis.

Clinically, AD is characterized by several features, notably a progressive cognitive decline involving loss of memory and higher executive functioning[37]. Excessive SA in mice, which was induced via KD of *Ska2*, resulted not only in severe hippocampal neuroinflammation and neurodegeneration, but also in cognitive impairment.

**Fig. 7 | Secretory autophagy is increased in human postmortem Alzheimer's disease samples. A** SNAP29 co-immunoprecipitation (SKA2 IP) and whole cell extract (WCE) control in amygdala (AMY), hippocampus (HIP), and prefrontal cortex (PFC) human postmortem samples (*n* = 3). **B** SKA2 immunostaining in neurons in CA1 stratum pyramidale (left) and microglia in CA1 stratum oriens (right) of the HIP from control subjects (*n* = 5). **C** Representative co-immunohistochemistry (IHC) image (left) and quantification (right) of SKA2 (red) and neuronal marker Camk2a (blue) in the HIP (*n* = 5). **D** Representative co-IHC image in stratum oriens CA1 of the HIP (left) and quantification (right) of SKA2 (red) and microglia marker IBA1 (blue) (*n* = 5). **E**–**I** Co-IHC images of the HIP from control subjects (*n* = 5 control subjects) depict cells labeled with antibodies against SNAP29, SKA2 or DAPI, and the overlap of the three markers. The colocalization of SKA2 with SNAP29 was observed on the cell-surface and cytoplasm of neurons (**F**) as well as in microglia (**G**–**I**). High-resolution images capturing single slices through the z-axis indicate that the majority of SKA2 and SNAP29 labeling is not localized within the DAPI-positive nuclei (H&I). **J** AD cohort from the Harvard Brain Tissue Resource Center (Ctrl *n* = 13 (8M/5F), AD *n* = 7 (4M/3F)): SKA2 protein expression (left) is significantly decreased in the hippocampus of AD subjects (ANCOVA: $F_{1,19}$ = 6.9123, *p* = 0.0170) while SEC22B binding to SNAP29 (right) is significantly increased in hippocampus tissue of AD subjects (ANCOVA: $F_{1,19}$ = 5.6769, *p* = 0.0284). **K** AD cohort from the Manchester Brainbank (Ctrl *n* = 37 (10M/27F), AD *n* = 40 (14M/26F)): SKA2 protein expression (left) is significantly decreased in the prefrontal cortex (PFC) of AD subjects (ANCOVA: $F_{1,76}$ = 6.4994, *p* = 0.0128). SEC22B binding to SNAP29 (middle) as well as NEK7 binding to NLRP3 (right) is significantly increased in PFC tissue of the top 12 low (5M/7F) compared to the top 12 high (5M/7F) SKA2 expressing AD subjects (ANCOVA: SEC22B to SNAP29: $F_{1,23}$ = 2.411, *p* = 0.0247, NEK7 to NLRP3: $F_{1,23}$ = 3.696, *p* = 0.0013). * = *p* < 0.05; data are presented as mean ± SEM. Scale bars represent 50 μm for (**B**, **D**), 100 μm for (**C**), and 20 μm for (**E**–**I**). Source data are provided as a Source Data file.

Intriguingly, PheWAS identified immunological and cognitive traits such as monocyte count, intelligence and cognitive performance with the *SKA2* locus as well as immunological phenotypes such as lymphocyte count with the *FKBP5* locus. Moreover, the *FKBP5* SNP variant rs1360780 has been associated with altered cognitive function in aged individuals[38].

Importantly, our human postmortem data suggest that markers of SA activity are increased in the hippocampus and PFC of AD brains (i.e., decreased SKA2 expression along with enhanced SEC22B to SNAP29 binding). Along these lines, increased FKBP5 expression has previously been linked to AD in several brain regions, and higher FKBP5 levels were associated with AD progression[39]. Human genetic studies have identified microglia as a key cell type governing the risk for AD[1,2]. Notably, *FKBP5* mRNA expression is increased in microglia of entorhinal cortex postmortem samples from individuals with AD[40]. Together, these results provide further evidence for the involvement of SA and its key regulators, SKA2 and FKBP5, in cognitive function and AD pathology.

There is increasing evidence from epidemiological and preclinical studies for the effects of environmental factors including early-life and chronic stress as well as traumatic experiences on microglia biology, which in turn might affect an individual's susceptibility to neurodegenerative diseases[2,41–43]. However, the underlying molecular mechanisms that mediate the crosstalk between neuronal stress circuits and the immune system remain largely unclear. Previous studies suggest that stress-exposure may precipitate disease risk by increasing inflammation in the periphery and in the brain[44–47]. Mechanistically, the effects of stress on neuroinflammation, and ultimately disease risk, could be mediated by stress-responsive genes and pathways able to modulate immune function. Indeed, the stress-inducible protein FKBP5 has been to shown to contribute to NF-κB-driven inflammation[48]. Notably, we have recently demonstrated that dexamethasone and glucocorticoid-mediated stress enhance SA via FKBP5, thereby driving extracellular BDNF maturation and synaptic plasticity as well as elevated immune signaling[7]. In the current study, our data reveal that SA-dependent and stress-induced release of IL-1β is impaired in the mPFC of *Fkbp5* KO mice. This puts SA in a prime position to mediate the crosstalk between neuronal stress circuits and the immune system. Thus, in the brain, depending on SA's activity level and specific cargo, this pathway might be involved in the entire spectrum of processes ranging from synaptic plasticity during learning and memory to neuroinflammation-induced neurodegeneration in the pathophysiology of diseases such as AD.

SKA2 expression was also shown to be regulated by glucocorticoids and involved in GR signaling. However, in contrast to FKBP5, SKA2 expression is decreased following dexamethasone treatment and SKA2 is suggested to enhance GR translocation to the nucleus in A549 human lung epithelial cells[11]. Thus, chronic or severe traumatic stress might lead to increased FKBP5 expression and decreased SKA2 levels, thereby increasing the activity of SA, which in the long run may precipitate in neurotoxicity and neurodegeneration.

Multiple lines of evidence support the pathogenic role of neuroinflammation in psychiatric illness. Elevated levels of central and peripheral cytokines have been detected in individuals with childhood trauma and stress-related psychiatric disorders[49–51]. Notably, single nucleotide polymorphism and epigenetic marks within the *FKBP5* and *SKA2* genes have repeatedly been associated with stress-related psychiatric diseases including major depressive disorder (MDD) and PTSD as well as suicide risk[12,13,15,18,52,53]. This is interesting considering that psychiatric illnesses such as MDD and PTSD can increase the risk for dementia and AD[54–56].

The current study also comes with a number of limitations. While we were able to demonstrate a direct protein-protein interaction between SKA2 and SNAP29, the identification of mutations affecting the SKA2-SNAP29 interaction remains an intriguing avenue for future exploration. This aspect should be addressed in forthcoming publications. Furthermore, the results obtained from microglia cell cultures along with the co-immunohistochemistry data of human microglia strongly support the role of SKA2 in IL-1β release and neuroinflammation. Currently, there are no cell type-specific *Ska2* knockout mouse lines available to further dissect the role of SKA2 in microglia in vivo. Future studies will be needed to further dissect the role of SKA2 in SA and neuroinflammation in a cell type-specific manner. It is worth mentioning that while we have focused on AD as an illustrative example, it may be plausible that SKA2, FKBP5, and the SA pathway may also play roles in other neurodegenerative diseases.

In summary, this study identifies SKA2 as a crucial molecular roadblock of SA in the mammalian brain. Our work highlights the central role of SA in the regulation of inflammasome activation and neuroinflammation-induced neurodegeneration, as well as its implication in the pathophysiology of AD.

# Methods

## Ethics statement

The research presented here complies with all relevant ethical regulations. All mouse experiments conformed to National Institutes of Health guidelines and were carried out in accordance with the European Communities' Council Directive 2010/63/EU and the McLean Hospital Institutional Animal Care and Use Committee. All efforts were made to minimize animal suffering during the experiments. The protocols were approved by the committee for the Care and Use of Laboratory animals of the Government of Upper Bavaria, Germany or by the local Institutional Animal Care and Use Committee, respectively. According to National Institutes of Health guidelines, research using de-identified postmortem tissue is not considered as human subject research. Approval for donation and use of the biological samples for postmortem analyses has been obtained from the Harvard Brain Tissue Resource Center / NIH NeuroBioBank (HBTRC/NBB) through the Mass

General Brigham IRB under protocol #2015P002028, and from the Manchester Brain Bank through the Newcastle & North Tyneside 1 Research Ethics Committee (19/NE/0242).

## Cell culture

**Neuro-2a cells.** N2a cells (ATCC, CCL-131) were maintained under standard conditions in Dulbecco's Modified Eagle Medium (DMEM) supplemented with 10% FBS and 1% antibiotic-antimycotic (all Thermo Fisher Scientific) at 37 °C and 5% CO$_2$ (vol/vol). For cell culture experiments, cells were seeded in 24 well plates at 35,000 cells/well. Transfection was performed the next day using Lipofectamine 2000 Transfection Reagent (Thermo Fisher Scientific), following the manufacturer's protocol. Cells were harvested 48 h post transfection using TrypLE Express (Thermo Fisher Scientific).

**SIM-A9 cells.** The murine microglia cell lines SIM-A9 wild type (Kerafast, END001), SIM-A9 *Sec22b* KO and SIM-A9 *Fkbp5* KO[7] were cultured at 37 °C, 6% CO$_2$ in DMEM high glucose with GlutaMAX (Thermo Fisher Scientific, 10566016), supplemented with 10% FBS (Thermo Fisher, 10270-106), 5% horse serum (Thermo Fisher Scientific, 16050-122) and 1% antibiotic-antimycotic (Thermo Fisher Scientific, 15240-062). With 1x trypsin-EDTA (Thermo Fisher Scientific, 15400-054) detached SIM-A9 cells (2 × 10$^6$) were resuspended in 100 µl of transfection buffer [50 mM HEPES (pH 7.3), 90 mM NaCl, 5 mM KCl, and 0.15 mM CaCl$_2$]. Up to 2 µg of plasmid DNA was added to the cell suspension, and electroporation was carried out using the Amaxa 2b-Nucleofector system (Lonza). Cells were replated at a density of 10$^5$ cells/cm$^2$.

## Treatment of SIM-A9 cells

Treatments of SIM-A9 cell cultures included PIK3C3/ Vps34 inhibition by SAR405 (0.1, 1 and 10 µM, 4 h; 533063, Merck), ULK1 inhibition by MRT68921 (0.1, 1 and 10 µM, 4 h; SML1644, Sigma-Aldrich) and stimulation of SA through the induction of lysosomal damage by LLOMe (0.25 mM, 4 h; L7393, Sigma-Aldrich). BafA1 (0.1 µM, 4 h; J61835, Thermo Scientific Chemicals) was used along with SAR405 or MRT68921 to asses autophagy flux and validate inhibition of autophagy flux via LC3 Western Blotting according to the guidelines for the use and interpretation of assays for monitoring autophagy[57].

## Autophagy flux assay in SIM-A9 cells

For autophagy flux assays, SIM-A9 cells were seeded in 12 well plates at a density of 200,000 cells/well. After 24 h, cells were treated with SAR405 or MRT68921 alone and with SAR405 or MRT68921 in combination with BafA1 at the indicated concentrations and time. For whole-cell lysate preparations, confluent cultures were washed three times with ice-cold DPBS (Thermo Fisher Scientific, 14190144) and lysed in an appropriate volume of SDS lysis buffer (50 mM Tris-HCl; 4855.2, Carl Roth; 1% SDS; 1057.1, Carl Roth; pH 8.0), freshly supplemented with 1x cOmplete™ EDTA-free protease inhibitor cocktail (4693132001, Roche) and 1× PhosSTOP™ phosphatase inhibitor cocktail (4906837001, Roche) for 20 min on ice. Subsequently, samples were prepared for separation with an SDS-PAGE system by the addition of 4× Laemmli sample buffer (1610747, Bio-Rad) containing 100 mM DTT (6908.1, Carl Roth) and heated at 95 °C for 5 min. Separation of proteins was performed with the Criterion™ Vertical Electrophoresis Cell System (1656001, Bio-Rad) on 4–20% Criterion™ TGX™ Precast Midi Protein Gels (5671095, Bio-Rad). Immunoblotting was performed with the Criterion™ Blotter (Bio-Rad, 1704070) on 0.2 µm PVDF membranes (1620177, Bio-Rad). For immunodetection, membranes were placed in TBS, supplemented with 0.05% Tween (P9416, Sigma-Aldrich) and 5% non-fat milk (T145.1, Carl Roth) for 1 h at RT and then incubated with the primary antibody diluted in TBS-T overnight at 4 °C. Following primary antibodies were used: LC3B (1:1,000, 2775, Cell Signaling Technology)

and GAPDH (1:1,000, 5174, Cell Signaling Technology). Then, immunoblots were washed and probed with horseradish peroxidase-conjugated secondary antibody (1:10,000, 7074, Cell Signaling Technology) for 1 h at RT. Immunoreactive protein bands were visualized using Clarity Western ECL Substrate (1705061, Bio-Rad) and determination of the band intensities was performed with ChemiDoc MP (12003154, Bio-Rad).

## Secretion assay in SIM-A9 cells

For cathepsin D secretion assays, SIM-A9 cells were seeded in 12 well plates at a density of 200,000 cells/well in a minimum volume of growth medium.

After 24 h, cells were treated with (1 or 10 µM) SAR405 or MRT68921 alone and with SAR405 or MRT68921 in combination with LLOMe (0.25 mM) at the indicated concentrations and time. Supernatants derived from cell cultures were collected, centrifuged, and subsequently stored at −80 °C until analysis. In capillary-based immunoassays using Jess (004-650, ProteinSimple, Bio-Techne), supernatants were separated on 12–230 kDa Separation Modules (SM-FL001, ProteinSimple, Bio-Techne). Cathepsin D antibody AF4G5 (LF-MA0321, Abfrontier) was used at a dilution of 1:50 and horseradish peroxidase-conjugated secondary antibody at 1:100 (7076, Cell Signaling Technology). Compass software was used for quantification of obtained signals (ProteinSimple, Bio-Techne).

## Animals & animal housing

Male mice, aged 2–4 months, were used for all experiments. For experiments in wild-type animals, C57BL/6J mice were obtained from The Jackson Laboratory (Bar Harbor, ME, USA). For in vivo brain microdialysis experiments, C57BL/6NCrl mice (Martinsried, Germany) as well as global *Fkbp5*$^{-/-}$ [58] mice and respective wild type controls (Martinsried, Germany) were used. Neonatal (P7-9) Thy1-GFP M mice (sex not determined) with a sparse expression of green fluorescent protein (GFP) in principal neurons in cortex and hippocampus, Jackson Laboratory Stock #007788[59] were used in organotypic hippocampal slice cultures (OHSC) experiments. All animals were kept under standard laboratory conditions and were maintained on a 12 h light–dark cycle (lights on from 0700 h to 1900 h), temperature 22 ± 1 °C, humidity 50%, in clear Plexiglas cages (19 × 29 × 13 cm$^3$) with unrestricted access to food (Purina Laboratory Rodent Diet 5001) and water.

## Preparation of organotypic hippocampal slice cultures (OHSC)

Neonatal Thy1-GFP M pups aged between P7-9 (sex not determined) were decapitated, brains removed, and hippocampi isolated from both hemispheres in ice-cold 1× minimum essential medium (MEM) with EBSS, 25 mM HEPES and 10 mM Tris buffer (pH 7.2) supplemented with Penicillin (100 IU/ml) and Streptomycin (100 µg/ml). Hippocampi were cut into coronal slices (thickness 350 µm) using a tissue chopper (McIlwain). Slices were transferred onto confettis Millipore biopore membrane, ~3 × 5 mm) which were placed on semiporous Millicell-CM inserts (0.4 µm pore size; Merck-Millipore). The inserts were put into cell culture dishes (35 mm, 4 slices/dish). OHSC were cultured according to the interface method[60] in 1 ml medium per dish at pH 7.2, 35 °C and a humidified atmosphere with 5% CO$_2$. Culture medium contained 0.5 × MEM with EBSS and 25 mM HEPES, 1 mM L-Glutamine, 25% Hanks' Balanced Salt solution (HBSS), 25% heat-inactivated horse serum, Penicillin (100 IU/ml) and Streptomycin (100 µg/ml; all Fisher Scientific). The medium was changed 1 day after preparation and every other day afterwards. Knockdown experiments were performed between 13 and 15 days in culture (DIC). Medium was harvested and snap frozen in liquid nitrogen directly. OHSCs were lysed in T PER™ Tissue Extraction Reagent (Thermo Fisher, 78510), supplemented with protease (Sigma, P2714) and phosphatase (Roche, 04906837001) inhibitor cocktails.

## Human postmortem studies

Tissue blocks were obtained from the Harvard Brain Tissue Resource Center/NIH NeuroBioBank (HBTRC/NBB), McLean Hospital, Belmont, MA, USA. *Healthy control subjects:* Tissue blocks containing the hippocampus, PFC (Brodmann area 9) and the amygdala from donors with no history of neurologic or psychiatric conditions were used for histochemical and immunocytochemical investigations as well as for immunoprecipitation analyses. See Supplementary Data 8, 9 for details of all subjects. All brains underwent a neuropathological examination including several brain regions. These studies did not include subjects with evidence for gross and/or macroscopic brain changes, or clinical history, consistent with cerebrovascular accident or other neurological disorders. Subjects with Braak stages III or higher (modified Bielchowsky stain) were not included. None of the subjects had significant history of substance dependence within 10 or more years from death, as further corroborated by negative toxicology reports. Absence of recent substance abuse is typical for samples from the HBTRC, which receives exclusively community-based tissue donations. Postmortem diagnoses were determined by two clinicians on the basis of retrospective review of medical records and extensive questionnaires concerning social and medical history provided by family members. *Alzheimer's disease discovery cohort:* Tissue blocks containing the hippocampus from donors with AD ($n = 7$, (4M/3F)) and healthy control subjects ($n = 13$, (8M/5F)) were used for western blotting and co-immunoprecipitation. All subjects were characterized clinically and neuropathologically as above. "Control" cases had Braak & Braak scores of 0-II, sparse plaque pathology, and were rated as having low probability of AD. AD cases had Braak & Braak scores of III-VI and were rated as having intermediate or high probability of AD. Neither group presented with additional relevant neuropathological findings. Groups were matched based on demographic factors (Supplementary Data 10).

*Alzheimer's disease replication cohort:* Fresh, frozen tissue was taken from the superior frontal gyrus (Brodmann area 8) of the frontal cortex from 77 brains (AD: $n = 40$ (14M/26F), Ctrl: $n = 37$ (10M/27F)) of donors who were participants of a large prospective cognitive aging cohort known as the University of Manchester Longitudinal Study of Cognition in Normal Healthy Old Age Cohort[61,62]. Samples were used for western blotting. Samples were acquired through the Manchester Brain Bank with ethical approval granted from the Manchester Brain Bank Committee. AD neuropathology was determined as described above. Groups were matched based on demographic factors (Supplementary Data 11).

## Plasmids

*shRNA Construction.* shRNA plasmids against mSka2 were constructed as follows: A shRNA plasmid containing a U6 promoter and a multiple cloning site followed by a mCherry gene driven by the PGK promoter was purchased from VectorBuilder Inc (Santa Clara, CA). Target sequences for mSka2 were derived from https://www.sigmaaldrich.com/life-science/functional-genomics-and-rnai/shrna/individual-genes.html. We designed custom 58nt oligos with AgeI/EcoRI restriction sites, annealed them to generate double stranded DNA fragments and ligated this fragment into the AgeI/EcoRI sites of pshRNA to generate Ska2-shRNA-1 (Ska2-shRNA-1: 5' CGAGAGGATCGTGATGCATT TCTCGAGAAATGCATCACGATCCTCTCG 3') and Ska2-shRNA-2 (Ska2-shRNA-2: 5' ACTGATACCCAGCATTCATTTCTCGAGAAATGAATGCTG GGTATCAGT 3'). Similar, a scrambled control was constructed (Scrambled-shRNA sequence: 5' CCTAAGGTTAAGTCGCCCTCGCTCG AGCGAGGGCGACTTAACCTTAGG 3'). Restriction digest and Sanger Sequencing confirmed the resulting plasmids.

*mSka2 overexpression.* The plasmid overexpressing Ska2 (EF1A>mSka2[NM_025377.3]:IRES:EGFP) and its control (EF1A > EGFP) were purchased from VectorBuilder Inc (Santa Clara, CA).

*Fkbp5 overexpression.* pRK5-FKBP5-FLAG have been described previously[63].

## Immunoblotting analysis

Frozen human brain tissue was pulverized on dry ice using a pre-cooled mortar and pestle, then transferred to an ice-cold homogenizer on ice. Mice were sacrificed by decapitation in the morning (08:00 am–08:30 am) following quick anesthesia by isoflurane. Brains were removed, snap-frozen in isopentane at −40 °C, and stored at −80 °C until further processing. Tissue punches of the PFC, hippocampus and amygdala were collected. Protein extracts from cell lines, mouse brains or pulverized human postmortem brains were obtained by lysing in T PER™ Tissue Extraction Reagent (Thermo Fisher, 78510) or lysis radio-immuno precipitation (RIPA) buffer, supplemented with protease (Sigma, P2714) and phosphatase (Roche, 04906837001) inhibitor cocktails. Samples were sonicated and heated at 95 °C for 10 min if necessary. Proteins were separated by SDS–polyacrylamide gel electrophoresis (PAGE) and electro-transferred onto nitrocellulose membranes. Blots were placed in Tris-buffered saline solution supplemented with 0.05% Tween (Sigma, P2287; TBS-T) and 5% non-fat milk for 1 h at room temperature and then incubated with primary antibody (diluted in TBS-T) overnight at 4 °C. Subsequently, blots were washed and probed with the respective horseradish-peroxidase- or fluorophore-conjugated secondary antibody for 1 h at room temperature. The immuno-reactive bands were visualized either using ECL detection reagent (Millipore, WBKL0500) or directly by excitation of the respective fluorophore. Recording of the band intensities was performed with the ChemiDoc MP and corresponding ImageLab software from Bio-Rad or the Odyssey CLx system interfaced with Image Studio version 4.0. Microdialysates obtained from in vivo acute stress experiments were analyzed using capillary-based immunoassays (Jess, ProteinSimple) and IL1B (1:50, Gene Tex, GTX74034) antibody.

*Quantification:* All protein data were normalized to ACTIN, GAPDH or VCP, which was detected on the same blot.

*Primary antibodies used:* FKBP5 (1:1000, Bethyl, A301-430A), FKBP5 (1:1000, Cell Signaling Technology, #12210), ACTIN (1:5000, Santa Cruz Biotechnology, sc-1616), GAPDH (1:8000, Millipore CB1001), SNAP29 (1:1000, Sigma, SAB1408650), SNAP23 (1:1000, Sigma, SAB2102251), STX3 (1:1000, Sigma, SAB2701366), STX4 (1:1000, Cell Signaling Technology, #67657), SEC22B (1:1000, Abcam, ab181076), SKA2 (1:1000, Thermo Fisher, PA5-20818), SKA2 (1:500, Millipore-Sigma, SAB3500102), SKA1 (1:1000, Thermo Fisher, PA5-20817), SKA3 (1:1000, Thermo Fisher, PA5-20819), GSDMD (1:1000, Cell Signaling Technology, 39754), VCP (1:10000, Abcam, Ab11433), NEK7 (1:50, Abcam, Ab133514), NLRP3 (1:50, Cell Signaling Technology, 15101), HIS (1:5000, Cell Signaling Technology, #12698), FLAG M2 (1:10000, Sigma, A8592).

*Secondary antibodies used:* anti-rabbit-IgG (1:1000, Cell Signaling, 7074), anti-mouse-IgG (1:1000, Cell Signaling, 7076), IRDyes 800CW donkey anti-Rabbit (1:20,000, LI-COR Biosciences, 926-32213), IRDye 680RD goat-anti-mouse (1:20,000, LI-COR Biosciences, 926-68070).

## Co-immunoprecipitation

For immunoprecipitation, cells were cultured for 3 days after transfection. Cells were lysed in Co-IP buffer [20 mM tris-HCl (pH 8.0), 100 mM NaCl, 1 mM EDTA, and 0.5% Igepal complemented with protease (Sigma) and phosphatase (Roche, 04906837001) inhibitor cocktail] for 20 min at 4 °C with constant mixing. The lysates (from cells or brain tissue as described above) were cleared by centrifugation, and the protein concentration was determined and adjusted (1.2 mg/ml); 1 ml of lysate was incubated with 2.5 µg of SEC22B, SNAP29 or SKA2 antibody overnight at 4 °C with constant mild rotating. Subsequently, 20 µl of bovine serum albumin (BSA)- blocked protein G Dynabeads (Invitrogen, 100-03D) were added to the lysate-antibody mix followed by a 3 h incubation at 4 °C. Beads were washed three times with phosphate buffered saline (PBS), and bound proteins were eluted by adding 60 µl of Laemmli sample buffer and by incubation at 95 °C for 5 min. 5–15 µg of the input lysates or 2.5–5 µl of the

immunoprecipitates were separated by SDS–PAGE and analyzed by western blotting. Immunoprecipitates of protein extracts obtained from human post mortem brains were analyzed using capillary-based immunoassays (Jess, ProteinSimple). When quantifying co-immunoprecipitated proteins, their signals were normalized to input protein and to the precipitated interactor protein.

## Pull-down assays

The pull-down was performed as described before in ref. 64. As bait protein, 200 ng either of HIS-SKA2 (Abcam, AB199577-1001) or Flag-FKBP51-His (provided by Thomas M Geiger and Felix Hausch, Technical University Darmstadt, Germany) was immobilized to bovine serum albumin (BSA)- blocked anti-His or anti-Flag-conjugated protein G Dynabeads (Invitrogen, 100-03D, HIS Cell Signaling Technology, #12698, FLAG M2, Sigma, F3165). Purified Syntaxin3-DDK (Origene, TP300658), SNAP29-DDK (Origene, TP302179), Syntaxin4 (Origene, TP300347), SNAP23-DDK (Origene, TP301596) or correspondingly SEC22B-HIS (Origene, AR50533PU-S) (100 ng) was used for the binding reaction. The binding reaction was carried out in a final volume of 400 µl of binding buffer (Promega, V870B). Elution was carried out by adding Laemmli buffer first at room temperature and then by boiling beads at 95 °C for 10 min.

## ELISA

The solid-phase sandwich ELISA (enzyme-linked immunosorbent assay) for the following antibody detection was performed according to the manufacturer's protocol: IL-1β (Thermo Fisher, BMS6002). Briefly, microwells were coated with mouse antibody followed by a first incubation with biotin-coupled anti mouse antibody, a second incubation with streptavidin-HRP and a final incubation with the SIM-A9 culture medium. Amounts of respective proteins were detected with a plate reader (iMARK, Bio-Rad) at 450 nm.

## RNA extraction and qPCR

Total RNA was isolated and purified using the Quick-RNA Miniprep Kit (Zymo Research, Irvine, CA) according to the manufacturer's protocol. RNA concentration was measured with The Qubit 2.0 Fluorometer (Thermo Fisher Scientific). RNA was reverse transcribed with the SuperScript IV First-Strand Synthesis System (Thermo Fisher Scientific, 18091050), using random hexamer primers provided within the kit. cDNA was amplified on an Applied Biosystems ViiA7 Real-Time PCR System with Power SYBR Green PCR Master Mix (Thermo Fisher Scientific, A25777). *Gapdh* was used as control. Data were analyzed using the $\Delta\Delta$Ct method unless otherwise stated. The following primer combinations were used: Ska2-fwd 5' CCAAGAGCTGCATTTGTGCT 3', Ska2-rev 5' GGCTCTGTTGCAGCTTTCTC 3'; Gapdh-fwd 5' TATGACTCC ACTCACGGCAA 3', Gapdh-rev 5' ACATACTCAGCACCGGCCT 3'.

## Surgical procedures and viral injections

Mice were deeply anesthetized with ketamine/dexdormitor (medetomidine) mixture and placed in a stereotaxic apparatus (David Kopf Instruments, Tujunga, CA, USA). For virus delivery a 33-gauge microinjection needle with a 10-µl syringe (Hamilton) coupled to an automated microinjection pump (World Precision Instruments Inc.) at 2 nl/ s was used. Coordinates in millimeters from bregma were as follows [in mm]: A/P -1.9, M/L ± 1.3, D/V -1.8 and -1.3. At the end of the infusion, needles were kept at the site for 5 min and then slowly withdrawn. The injection volume was 0.5 µl. After bilateral infusion, incisions were sutured closed using nylon monofilament (Ethicon). During surgery, body temperature was maintained using a heating pad. After completion of surgery, anesthesia was reversed using Antisedan (atipamezole) and mice were allowed to recover on heating pads.

Surgeries for guide cannula implantations (microdialysis) were performed as previously described[7,65]. Coordinates for microdialysis probe guide cannula implantations into the right mPFC were (with

bregma as a reference point) as follows [in mm]: A/P 2.00, M/L 0.35, and D/V -1.50. After guide cannula implantation, animals were allowed to recover for 7 days in individual microdialysis cages (16 × 16 × 32 cm³). Metacam (0.5 mg/kg, s.c) was injected within the first 3 days after surgeries, when required.

## Microdialysis

The perfusion setup consisted of lines comprised of FET tubing of 0.15 mm ID (Microbiotech Se, Sweden), a 15 cm-PVC inset tubing (0.19 mm ID), a dual-channel liquid swivel (Microbiotech Se, Sweden). Perfusion medium was sterile RNase free Ringer's solution (BooScientific, USA) containing 1% BSA (Sigma, A9418). Perfusion medium was delivered to the probe at the flow rate of 0.38 µl/ min with the syringe pump (Harvard Apparatus, USA) and withdrawn with the peristaltic pump MP2 (Elemental Scientific, USA) at the flow rate of 0.4 µl/ min. Microdialysis CMA 12 HighCO Metal Free Probe was of 2 mm length membrane with 100 kDa cut off (Cat.N. 8011222, CMA Microdialysis, Sweden). All lines were treated with 5% poly-ethylenimine (PEI) for 16 h and then with $H_2O$ for 24 h before the experiments. The microdialysis probe was inserted into the implanted guide cannula (under 1–1.5 min isoflurane anesthesia, 2% v/v in air) 6 days after the stereotaxic surgery and 18 h before the samples collection. A baseline sample collection phase (three samples) was always preceding the foot shock (FS), which allowed us to express the changes in the extracellular content of proteins as relative to the baseline values. On the experimental day, microdialysis fractions were constantly collected (for 30 min) on ice into 1.5 ml Protein LoBind tubes (Eppendorf) preloaded with 0.5 µl protease inhibitor cocktail 1:50 (Roche) at a perfusion rate of 0.4 µl/min. After collection, samples were immediately frozen on dry ice for subsequent analysis. After collection of three baseline samples animals were transferred to the FS chamber (ENV-407, ENV-307A; MED Associates, 7 St Albans, VT, USA) connected to a constant electric flow generator (ENV-414; MED Associates) and a FS (1.5 mA × 1 s × 2) was delivered. After this procedure, mice were returned to the microdialysis cage where two post-FS samples were collected. To examine an effect of ULK1 inhibitor MRT68921 on stress-evoked changes in extracellular content of proteins, the drug was injected intraperitoneally at a dose of 5.0 mg/kg and in a volume 10 ml/kg 4 h before the FS (the drug was prepared freshly dissolving a stock solution [60% EtOH/40% DMSO mixture] with saline at 1:20). At the end of the experiment, probes were removed, brains were frozen and kept at -80 °C for the probe placement verification. 40 µm brain sections were stained with cresyl violet (Carl Roth GmbH) and probe placement was verified under a microscope. If probe placement was found to be out of the targeted region of interest, the respective samples were excluded from the study.

## Behavior

All experiments were analyzed using the automated video-tracking system ANYmaze (Stoelting, Wood Dale, IL).

**Open field (OF) test.** The OF test was used to characterize locomotor activity in a novel environment. Testing was performed in an open field arena (50 × 50 × 50 cm) dimly illuminated (10 lux) in order to minimize anxiety effects on locomotion. All mice were placed into a corner of the apparatus at the beginning of the trial. The testing duration was 10 min and the distance traveled was assessed.

**Y-maze.** The Y-maze test was used to assess spatial recognition memory. The apparatus consisted of three evenly illuminated arms (30 × 10 × 5 cm, 10 lux) with an angle of 120° between each arm. The apparatus was surrounded by various spatial cues. To ensure that the mice had sufficient spatial cues available without having to stretch up and look outside of the maze, we also introduced intra-maze cues (triangles, bars, and plus signs) that served the same purpose as the

external cues. The Y-maze test comprises two trials separated by an intertrial interval (ITI) to assess spatial recognition memory. During the first trial, the mouse was allowed to explore only two of the three arms for 10 min while the third arm was blocked. After a 30 min ITI, the second trial was conducted during which all three arms were accessible for 5 min and the time spent in each arm was assessed. An increased amount of time spent in the novel arm, relative to the familiar arms, reflects intact spatial recognition memory.

**Novel object recognition memory task.** Novel object memory was assessed in the Y-maze under low illumination (10 lux). During the acquisition trial, mice were presented with two identical objects and allowed to freely explore the objects for 10 min. Following a 30 min ITI, mice were presented with one familiar object and a novel one. During the retrieval phase mice were allowed to explore the objects for 5 min. At the start of each trial, mice were placed in the arm without an object. All objects were built from 10 LEGO bricks to allow a consistent volume, while shape and color could be varied to create distinguishable objects. The type of object that was chosen as familiar or novel respectively as well as the relative positions of the novel and familiar objects were counterbalanced across the groups. The time spent interacting with the objects was assessed and the ratio of time exploring the novel to the familiar object was calculated. A higher preference for the novel object reflects intact object recognition memory.

**Elevated plus maze (EPM).** The EPM was employed to assess anxiety-related behavior under low illumination (10 lux). The apparatus consisted of a plus-shaped platform with four intersecting arms, elevated 70 cm above the floor. Two opposing open ($30 \times 5$ cm) and closed ($30 \times 5 \times 15$ cm) arms were connected by a central zone ($5 \times 5$ cm). Animals were placed in the center of the apparatus facing the closed arm and were allowed to freely explore the maze for 5 min. Open arm time was calculated as a percentage of time in seconds: open arm time [%] = open arm time/(open arm time + closed arm time).

**Immunohistochemistry (mouse brain tissue)**
Mice were deeply anesthetized with isoflurane and perfused intracardially with PBS followed by 4% paraformaldehyde. Brains were removed, post-fixed overnight in 4% paraformaldehyde followed by an additional overnight incubation in 30% sucrose solution at 4 °C, and then stored at −80 °C. Frozen brains were coronally sectioned in a cryostat microtome at 35 μm. Slices were subsequently washed with PBS and blocked using 10% normal donkey serum prepared in PBS containing 0.1% Triton X-100 for 1 h at room temperature. Next, slices were incubated with the appropriate primary antibody (anti-NeuN, 1:1000, Synaptic Systems, 266004; anti-IBA1, 1:1000, FUJIFILM Cellular Dynamics, 019-19471; anti-GFAP, 1:1000, Cell Signaling Technology, 12389; anti-ASC, 1:200, AdipoGen, AG-25B-0006-C100; anti-CASP-1, 1:1000, Santa Cruz, sc-56036; anti-mCherry, 1:1000, Millipore Sigma, AB356481) in 10% NGS PBS overnight at 4 °C on a shaker. Then slices were washed three times (10 min each) with PBS and incubated with the Alexa Fluor 488 and 594 conjugated secondary antibodies (1:1000, Invitrogen, A-21206, A-11042, A-11076) in 10% NGS PBS for 2 h at room temperature. Following three washes (15 min each) with PBS, slices were mounted on superfrost plus slides and covered with Vectashield mounting medium (Vector Laboratories, Burlingame, USA) containing DAPI. Slides were stored at 4 °C until imaging.

**Imaging and quantification**
Sixteen-bit images were acquired on a Leica SP8 confocal microscope with 10× or 40× objectives at identical settings for all conditions. Images were quantified using ImageJ (https://imagej.nih.gov/ij). For each experimental condition, one to two coronal sections per mouse from the indicated number of animals were used.

*NeuN CA1 thickness.* NeuN staining was used to measure the CA1 thickness with ImageJ. Leica SP8 with a 10× objective was used to acquire the images. The identical portion of the dorsal hippocampus was imaged for each brain.

*Microglia.* IBA1 immunoreactive cells were considered microglia. Leica SP8 with a 10× objective was used to acquire the images. The identical portion of the dorsal hippocampus was imaged for each brain. The cell counter plugin in ImageJ was used to count cells manually. When microglia density was too high to count individual cells, the signal intensity was measured in ImageJ instead.

*Astrocytes.* GFAP immunoreactive cells were considered astrocytes. Leica SP8 with a 10× objective was used to acquire the images. The cell counter plugin in ImageJ was used to count cells manually. When astrocyte density was too high to count individual cells, the signal intensity was measured in ImageJ instead.

*CASPASE-1.* Leica SP8 with a 10× objective was used to acquire the images. CASP-1 signal intensity was measured in ImageJ.

*ASC.* Leica SP8 with a 40× objective was used to acquire the images. The cell counter plugin in ImageJ was used to count ASC+ cells as well as ASC specks manually.

**Analysis of ASC specks in SIM-A9 cells**
SIM-A9 wild type and SIM-A9 $Sec22b^{-/-}$ cells stably expressing ASC-mCerulean were used as reporter cells for inflammasome activation and generated as described before in ref. 24. For imaging experiments, SIM-A9 wild type and SIM-A9 $Sec22b^{-/-}$ cells expressing ASC-mCerulean were plated at a density of $2 \times 10^5$ cells/well on black 96-well plates (μ Plate, ibidi, Gräfelfing, Germany). Transfection of 200 ng of SCR- and Ska2-shRNA constructs was performed using Lipofectamine 3000 (Thermo Fisher Scientific) according to the manufacturer's instructions. 48 h post transfection, cells were stimulated with 200 ng/ml LPS from *E. coli* 026:B6 (Thermo Fisher Scientific) for 2 h and subsequently fixed using 4% PFA. Images for assessment of ASC specks in PFA-fixed SIM-A9 cells were acquired using the VisiScope CSU-W1 spinning disk confocal microscope and the VisiView Software (Visitron Systems GmbH). Settings for laser and detector were maintained constant for the acquisition of each image. For analysis, at least seven images were acquired using the 20x objective. For quantification of ASC specks, mCerulean signal resembling an ASC speck/cell was counted manually in ImageJ and normalized to the number of DAPI- or DRAQ5-positive nuclei (ratio to cell count).

**Production of adeno-associated viruses (AAVs)**
Packaging and purification of pAAV9-U6-shRNA[Ska2#1]-PGK-mCherry and pAAV9-U6-shRNA[Scr]-PGK-mCherry was conducted by Vigene Biosciences (Rockville, MD, USA). AAV9 titers were >$1 \times 10^{13}$ GC/ml. Packaging and purification of pAAV5-U6-shRNA[Ska2#2]-PGK-mCherry and pAAV5-U6-shRNA[Scr]-PGK-mCherry was conducted by the Viral Vector Core of Emory University (Altanta, GA, USA). AAV5 titers were >$1 \times 10^{11}$ GC/ml.

**RNA sequencing**
Mice were sacrificed 2 and 4 weeks after viral-mediated knockdown of *Ska2* (*n* = 8 per condition (Scr-shRNA, Ska2-shRNA) and timepoint (2 weeks, 4 weeks)). Brains were removed, snap-frozen in isopentane at −40 °C, and stored at −80 °C until further processing. Tissue punches of hippocampus were collected. Total RNA was isolated using the Agilent absolutely RNA miniprep kit. Samples were processed by mRNA enrichment and rRNA removal. The cDNA library was sequenced using the BGISEQ-500 platform by BGI (Hong Kong, China). 100 bp paired-end sequencing was done, which yielded on average 49.3 million raw reads of which on average 48.3 million (98%) mapped to the *M. musculus* (GRC38/mm10) genome using bcbio-nextgen (v.1.2.0)[66] with STAR[67] as aligner and Salmon[68] was used to quantify the

expression of the transcripts. Next, tximport[69] was used to create the gene-level count matrix from the Salmon quantifications. Quality control (QC) of the raw reads was carried out with MultiQC (v.1.8) to determine whether the sequencing data was suitable for subsequent analysis[70]. For additional QC Principal component analysis (PCA) was used.

### RNA sequencing analyses

**Cell type deconvolution.** Deconvolution of the bulk RNA-seq data leads to an estimated proportion of the different cell types in the RNA-seq analysis. Deconvolution was carried out using MuSiC (v.1.0.0)[28], with previously published single-cell RNA sequencing (scRNA-seq) data from adult male mice of the same age and strain[71]. In total 23.179 hippocampal cells were used from the scRNA-seq experiment to deconvolve the bulk RNA-seq data including the following cell types: Neurons, Astrocytes, Microglia/Macrophage, Interneurons, cells related to Neurogenesis, Oligodendrocytes, Polydendrocytes, Endothelial cells, Fibroblasts, Choroid plexus cells, Mural cells and cells from the Ependyma.

**Differential expression analysis.** All analyses were done in R 4.1.2 (unless stated otherwise). Raw data is filtered with that a gene needs at least 10 or more counts. Differential expression analysis was carried out on filtered read counts for genes by using DESeq2 (v.1.34.0)[72], using condition as the main predictor, and batch and proportion microglia cells as covariate. We computed differentially expressed genes (DEGs) by contrasting the control group vs the shRNA-Ska2 group for the 2 and 4 weeks timepoint. P-values were adjusted with the Bonferroni correction. The alpha threshold was $< 0.05$.

**Gene Ontology (GO) enrichment analysis.** GO enrichment analysis was carried out using clusterProfiler (v.4.2.2)[73] and org.Mm.eg.db (v.4.0) for annotation[74]. The DEGs (2 and 4 weeks) were used as input for GO enrichment analyses, and GO enriched terms were corrected with a Bonferroni p-value correction. Terms with an adjusted $p < 0.05$ were deemed as significantly enriched. Z-scores are calculated using GOplot (v.1.0.2)[75] which takes the logFC and ID per gene as input next to the output of the GO enrichment analysis.

**Kyoto Encyclopedia of Genes and Genomes (KEGG) enrichment analysis.** KEGG enrichment analysis was carried out using msigdbr (v.7.5.1) and clusterProfiler (v.4.2.2). KEGG enrichment analyses were corrected with a Bonferroni p-value correction. Pathways with an adjusted $p < 0.05$ were deemed as significantly enriched. Z-scores are calculated using GOplot (v.1.0.2)[75] which takes the logFC and ID per gene as input next to the output of the KEGG enrichment analysis.

### Immunohistochemistry (human brain tissue)
Tissue blocks for immunohistochemistry were dissected from fresh brains and post-fixed in 0.1 M phosphate buffer (PB) containing 4% paraformaldehyde and 0.1 M NaN$_3$ at 4 °C for 3 weeks, then cryoprotected at 4 °C (30% glycerol, 30% ethylene glycol and 0.1% NaN$_3$ in 0.1 M PB), embedded in agar, and pre-sliced in 2.5 mm coronal slabs using an Antithetic Tissue Slicer (Stereological Research Lab., Aarhus, Denmark). Each slab was exhaustively sectioned using a freezing microtome (American Optical 860, Buffalo, NY). Sections were stored in cryoprotectant at −20 °C. Sections were cut at 50 μm thickness through the hippocampus and collected in compartments in serial sequence. Four to six sections within one compartment/subject were selected for immunolabeling.

**Immunocytochemistry.** Antigen retrieval was carried out by placing free-floating sections in citric acid buffer (0.1 M citric acid, 0.2 M Na$_2$HPO$_4$) heated to 80 °C for 30 min. Sections were then incubated in primary antibody (SKA2, SAB3500102, lot#54031701, MilliporeSigma,

St. Louis, MO) for 48 h at 4 °C and then in biotinylated secondary serum (SKA2, goat anti-rabbit IgG 1:500; Vector Labs, Inc. Burlingame, CA). This step was followed by streptavidin conjugated with horseradish peroxidase for two h (1:5000, Zymed, San Francisco, CA), and, finally, nickel-enhanced diaminobenzidine/peroxidase reaction (0.02% diaminobenzidine, Sigma-Aldrich, 0.08% nickel-sulfate, 0.006% hydrogen peroxide in PB). All solutions were made in PBS with 0.5% Triton X unless otherwise specified. All sections were mounted on gelatin-coated glass slides, coverslipped, and coded for quantitative analysis blinded to age. Sections from all brains included in the study were processed simultaneously within the same session to avoid procedural differences. Each six-well staining dish contained sections from normal control subjects and was carried through each step for the same duration of time, so to avoid sequence effects. Omission of the first or secondary antibodies did not result in detectable signal.

**Dual antigen immunofluorescence.** Antigen retrieval as described above. Sections were co-incubated in primary antibodies (rabbit anti-SKA2, 1:300, SAB3500102, MilliporeSigma, St. Louis, MO; mouse anti-IBA1, 1:500, cat# 013-27593, Wako FujiFilm Chemicals USA Corp., Richmond, VA; mouse anti-CamKIIα 1:500, ab22609, Abcam, Cambridge, MA; mouse anti-SNAP29 1:100, sc-390602, Santa Cruz Biotechnology, Dallas, TX) in 2% BSA for 72 h at 4 °C. This step was followed by 4 h incubation at room temperature in Alexa Fluor goat anti-mouse 647 (1:300; A-21235, Invitrogen, Grand Island, NY) and donkey anti-rabbit 488 (1:300; A-21206, Invitrogen, Grand Island, NY) or goat anti-rabbit 555 (1:300, A-32732, Invitrogen, Grand Island, NY), followed by 10 min in DAPI (1:10,000, D1306, Invitrogen, Grand Island, NY) to label cell nuclei and 1 min incubation in TrueBlack solution (cat# 23007, Biotum Inc., Fremont, CA) to block endogenous lipofucsin autofluorescence[76]. Sections were mounted and coverslipped using Dako mounting media (S3023, Dako, North America, Carpinteria, CA).

**Data collection.** An Olympus BX61 interfaced with StereoInvestigator v.2019 (MBF Biosciences, Williston, VT) was used for analysis. The borders of the hippocampal subregions were identified according to cytoarchitectonic criteria as described in our previously published studies[77,78]. A 1.6x objective was used to trace the borders of hippocampal subregions. Each traced region was systematically scanned through the full x, y, and z axes using a 40x objective to count each immunoreactive (IR) cell within the traced borders over all sections from each subject.

Confocal imaging: A Zeiss LSM 880 confocal microscope interfaced with Zen imaging software (ZEN 2.3 SP1) was used to acquire images of SKA2 and SNAP29 cells. Images were acquired with a z-step of 0.5 μm using a 63x oil immersion objective (numerical aperture 1.4 DIC M27; pixel size, 0.10 × 0.10 μm). Maximum intensity projections were acquired using Zen Blue (ZEN 2.6) software.

**Numerical densities and total numbers estimates.** Numerical densities (Nd) were calculated as $Nd = \Sigma N / \Sigma V$ where $N$ is the sum of cells within a region of interest, and $V$ is the total volume of the region of interest as described previously in detail[79].

### PheWAS
Phenotypic data for the *FKBP5* and *SKA2* genes were obtained from the Atlas of GWAS Summary Statistics (http://atlas.ctglab.nl/PheWAS), database release3: (v20191115)[29].

### Statistics and reproducibility
Data are presented as means + standard error of the mean (SEM). Cell culture and mouse data were analyzed using GraphPad 10.0 (La Jolla, CA). When two groups were compared, paired or unpaired, two-tailed Student's t-test was applied, as appropriate. For three or more group comparisons, one-way, two-way or repeated measures two-way

analysis of variance (ANOVA) was performed, followed by Tukey's, Bonferroni Dunnett's or Šidák's multiple comparison post hoc test, as appropriate. JMP Pro v. 14 SW (SAS Institute Inc., Cary, NC) was used for analysis of covariance (ANCOVA) of human postmortem data. Differences between groups relative to the main outcome measures in each of the regions examined were assessed for statistical significance using an ANCOVA stepwise linear regression process. Age, sex, and postmortem time interval, hemisphere, and brain weight are included in the model if they significantly improved the model goodness-of-fit. *P* values of <0.05 were considered statistically significant. Experiments were informed by prior research indicating $n = 3$–7 biological replicates were sufficient for detecting significant differences for effect size estimations. For behavioral experiments, animal numbers were based on historical data, using G*Power 3 for estimation.

In the behavioral analysis of the novel object recognition test, one mouse of the SKA2-KD group (Ska2-shRNA) was excluded based on Grubbs' outlier test. For the EPM behavioral data, one mouse of the ctrl group (Scr-shRNA) fell off the open arm during testing and was excluded from the analysis. No other data were excluded. Experimental groups were assigned in a semi-randomized manner to ensure a balanced distribution across treatments. This initial randomization was deemed sufficient, as parallel processing of samples minimized potential variability. For AD and control human postmortem samples, selection was based on matching co-variates (age, sex, postmortem interval, brain hemisphere, and brain weight) and Braak & Braak staging for disease progression. All experiments and data analyses were completed by an experimenter blinded to the group allocation during data collection.

### Reporting summary
Further information on research design is available in the Nature Portfolio Reporting Summary linked to this article.

## Data availability
The *M. musculus* genome (GRC38/mm10) used to map the raw reads of the RNA sequencing data is available at: https://www.ncbi.nlm.nih.gov/datasets/genome/GCF_000001635.20/. The RNA sequencing data following *Ska2* knockdown (Fig. 6, Figs. S5, S6) generated in this study are available at Gene Expression Omnibus (GEO): GSE181203. Single-cell RNA sequencing data used for the deconvolution analysis are available at GEO: GSE116470. Original data for Figure S7A, B (Phenotypic data for the *FKBP5* and *SKA2)* genes were publicly available from the Atlas of GWAS Summary Statistics and can be downloaded at http://atlas.ctglab.nl/PheWAS. All unique materials used are readily available from the authors or standard commercial sources. Source data are provided with this paper.

## Code availability
Code for data cleaning and analysis of the RNA sequencing is available from GitHub at https://github.com/klengellab/Ska2.

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

## Acknowledgements

We thank all lab members for suggestions and comments on the experiments and manuscript. N.C.G. was supported by Stiftung Charité. This study was supported by a NARSAD Young Investigator Grant from the Brain & Behavior Research Foundation, honored by P&S Fund (awarded to N.C.G., Grant ID 25348), the National Institutes of Health (P50-MH115874 to W.A.C.jr/K.J.R., and R01-MH117292 to K.J.R). S.Mackert was funded by Volkswagen Foundation (awarded to N.C.G. Grant ID 9A889). T.K. was supported by research grants from NICHD (R21HD088931, R21HD097524, R01HD102974), NIMH (R21MH117609), and ERA-Net Neuron (01EW2003). Human tissue was obtained from the NIH NeuroBioBank. In addition, tissue samples were supplied by The Manchester Brain Bank, which is part of the Brains for Dementia Research programme, jointly funded by Alzheimer's Research UK and Alzheimer's Society. Figure 2C, D, parts of Fig. 3A and, Fig. 4L were created with BioRender.

## Author contributions

J.H., K.J.R., and N.C.G. conceived the project and designed the experiments. T.B., C.K., T. E., E.J., K.H., S.Mackert, T.M.G., T.R., S.Martinelli, F.H., and N.C.G. performed protein and cell culture experiments. J.H., C.K., A.K.G., E.A.A., F.T., G.M., M.L.P., D.E.H., N.D, K.M.M., T.P., V.S., E.L., W.A.C. and M.V.S. performed animal experiments. J. H., J.O., C. K., R. L., and T.K. performed and analyzed the RNA sequencing experiment. J.H., C.K., L.R., D.T.T., K.L., A.P., A.C.R., C.M., S.B., T.K., H.P., and N.C.G. performed human postmortem experiments. J.H., K.J.R., and N.C.G. wrote the initial version of the manuscript. J.H., K.J.R., and N.C.G. supervised the research. All authors contributed to the final version of the manuscript.

## Competing interests

N.D. is currently an employee of Sumitomo Pharma America. K.M.M. is currently an employee of Jazz Pharmaceuticals. S.M. is currently an employee of Roche Diagnostics. Their contributions pre-date that employment and their co-authorship do not reflect collaboration on this project with these organizations. K.J.R. has received consulting income from Alkermes, Bionomics, and BioXcel and is on scientific advisory boards for Janssen and Verily for unrelated work. He has also received a sponsored research grant support from Takeda, Alto Neuroscience, and Brainsway for unrelated work. T.K. has received consulting income from Alkermes for unrelated work. Within the last 2 years, W.A.C. has served as a consultant for Psy Therapeutics and has had sponsored research agreements with Cerevel Therapeutics and Delix Therapeutics. The remaining authors declare no competing interests.

## Additional information

¹Department of Psychiatry, Harvard Medical School, McLean Hospital, Belmont, MA 02478, USA. ²Research Group Neurohomeostasis, Department of Psychiatry and Psychotherapy, University of Bonn, 53127 Bonn, Germany. ³Department of Psychiatry and Psychotherapy, University Medical Center Göttingen, 37075 Göttingen, Germany. ⁴Department of Psychiatry and Psychotherapy, University of Bonn, 53127 Bonn, Germany. ⁵Department of Translational Research in Psychiatry, Max Planck Institute of Psychiatry, 80804 Munich, Germany. ⁶Research Group Neuronal Plasticity, Max Planck Institute of Psychiatry, 80804 Munich, Germany. ⁷Research Group Neurobiology of Stress Resilience, Max Planck Institute of Psychiatry, 80804 Munich, Germany. ⁸Department of Psychiatry and Human Behavior, University of Mississippi Medical Center, Jackson, MS 39216, USA. ⁹Institute for Organic Chemistry and Biochemistry, Technische Universität Darmstadt, 64287 Darmstadt, Germany. ¹⁰Institute of Innate Immunity, University Hospital Bonn, 53127 Bonn, Germany. ¹¹Division of

Informatics, Imaging and Data Sciences, University of Manchester, Manchester M13 9PL, UK. [12]Division of Neuroscience, Faculty of Biology, Medicine and Health, School of Biological Sciences, The University of Manchester, Salford Royal Hospital, Salford M6 8HD, UK. [13]Geoffrey Jefferson Brain Research Centre, Manchester Academic Health Science Centre (MAHSC), Manchester, UK. [14]Institute of Physiology II, University of Bonn, 53127 Bonn, Germany. [15]Deutsches Rheuma Forschungszentrum Berlin (DRFZ), 10117 Berlin, Germany. [16]Department of Life Sciences, Manchester Metropolitan University, Manchester M15 6BH, UK. [17]These authors contributed equally: Kerry J. Ressler, Nils C. Gassen. ✉e-mail: jhartmann@mclean.harvard.edu; kressler@mclean.harvard.edu; nils.gassen@ukbonn.de

