## [Peer Review File · Nature Communications]

SKA2 regulated hyperactive secretory autophagy drives neuroinflammation-induced neurodegenerationREVIEWER COMMENTS

Reviewer #1 (Remarks to the Author):

In this paper by Hartmann entitled “SKA2 regulated hyperactive secretory autophagy drives neuroinflammation induced neurodegeneration” the authors present data to support a model where secretory autophagy (SA) regulates neuroinflammation-mediated neurodegeneration via SKA2 and FKBP5. The latter stimulates SA and release of interleukin 1beta (IL-1b) whereas SKA2 inhibits SA-dependent IL-1 β release by counteracting FKBP5.

This is a very interesting paper with potentially very important findings. However, at the present stage I think that the conclusions are not completely validated by the data shown. Firstly, we need to be more convinced about how SKA2 and FKBP5 mechanistically act in SA. Secondly, is the dramatic effect of SKA2 knockdown (KD) on the hippocampus actually due to neuroinflammation caused by unchecked SA?

Major points:

1. Direct protein-protein interaction data are missing to strengthen the model of SKA2 and FKBP5 regulation of the SNARE complexes involved in secretory autophagy. Experiments testing out direct interactions and locating mutations that can compromise the interaction will strengthen this model. Does FKBP5 bind directly to SEC22B? Does SKA2 bind to STX3 or -4 or both or to SNAP29 or -23 or both? This needs to be tested.
2. Another missing link is colocalization studies showing that SKA2 and FKBP5 colocalize with the relevant components in the SA complexes mentioned in 1.
3. Fig 2A: Following viral-mediated shRNA-dependent KD of Ska2 in the hippocampus of C57Bl/6J mice complete hippocampal atrophy occurred within six weeks. As its name implies, spindle and kinetochore-associated complex subunit 2 (SKA2) has a known function as a scaffolding protein forming the SKA complex together with SKA1 and SKA3. This complex is a key component of the kinetochore-microtubule interface. The SKA complex is required for normal regulation of cell cycle checkpoint. KD of SKA2 expression by RNAi is known to block cell cycle progression during metaphase. Cells can often complete mitosis, but it is delayed, kinetochore fibers are destabilized and there is often a failure of spindle assembly checkpoint exit (EMBO J. 25, 5504, 2006). Mutations that disrupt the SKA complex can lead to cell death (Molecular Cell DOI 10.1016/j.molcel.2012.03.005).

The authors conclude from their KD SKA2 experiments in the hippocampus that: “Together, these data provide significant mechanistic evidence that hyperactivated SA (through KD of Ska2) is able to create an inflammatory feed-forward vicious cycle resulting in a GSDMD-mediated excessively neurotoxic environment to ultimately catalyze neuroinflammation and neurodegeneration.” The question is if this now occurs secondarily to cell death caused by SKA2 KD or if there is a direct effect on secretory autophagy of IL1b by removal of SKA2 as a negative regulator of SA? Hence, can the authors show that the effects they see is not due to apoptosis or other type of cell death? The neuroinflammatory effects

may be secondary to the cell death.

4. The known roles of SKA2 are not presented in the paper at all which the authors certainly should as part of either the INTRODUCTION or DISCUSSION, or both. Can the authors rationalize some of the known roles with the role they suggest SKA2 has in secretory autophagy?

5. Does SKA1 or SKA3 have any role in SA? Is SKA2 acting alone in SA without its partner proteins in the SKA complex?

6. Has the SKA2 Ab used for IP been validated for specificity? It would have been nice to see a negative control in the IP experiment in Fig. 1A where all tested proteins are positive.

7. The authors employ in Fig. 1R the ULK1 inhibitor (ULK1i) MRT68921 to show that autophagy is involved in the secretion of IL1 β . This experiment is very important and as it stands it rests on the specificity of this inhibitor for ULK1. The problem is that we cannot rule out TBK1 and AMPK which both are inhibited by MRT68921 and both of them are involved in autophagy (PMID: 25833948). The authors should also do this experiment using a VPS34 inhibitor like SAR405.

Minor points:

8. The blot in Fig 1H shows a band at the expected MW for FKBP5 in the extract from FKBP5 KO. Is the KO not complete, or is it the ab recognizing both FKBP51 and FKBP52 or another protein?

9. In The Abstract on line 49 the expression “The SKA2-mediated hyperactivation of SA” is not a correct expression as SKA2 is actually knocked down and as such not “mediating” it is rather the loss of SKA2 activity that mediates or leads to hyperactivation of SA.

10. Line 139 “1” is missing in “MRT68921”

Reviewer #2 (Remarks to the Author):

This study by Hartmann et al attempts to reveal the role of SKA2-secretory autophagy (SA) pathway in inflammation-mediated neurodegeneration. The authors identified that SKA2 and FKBP5 had the opposite effects on RQ-SNARE complex formation and the subsequent IL-1 β release. To further investigate the role of SKA2 in the regulation of neuroinflammation, AAV was injected into the mouse hippocampus to knock down Ska2, which results in microglia activation, inflammasome formation, hippocampal atrophy and cognitive impairment. Detection of postmortem brains from Alzheimer's patients found the reduction of SKA2 level and increase of secretory autophagy (SA). Overall, it is an interesting study to identify a novel mechanism of regulating SA-dependent neuroinflammation. However, there are some major concerns about the study design and data analysis. The mechanistic studies should be further strengthened to support the conclusion.

Major concerns:

1. Although the author used postmortem brain samples of AD patients, there is no evidence from any cellular or animal study to show the link of SKA2-SA pathway with any pathological hallmarks of AD, such as A β plaques, tau phosphorylation and neuronal death. Without these data, the role of SKA2 in the pathogenesis of AD cannot be confirmed.

2. The results of figure 1 show the interactions among SKA2, FKBP5 and the components of SNARE complex, but the underlying mechanisms are not clear. How do SKA2 and FKBP5 regulate each other (Figure 1B & H)? Which component of SNARE complex directly binds with SKA2? How does FKBP5 regulate the binding of SEC22B with SNAP29? It seems that FKBP5 regulates the expression of SEC22B (Figure 1E). Whether this is the mechanism by which overexpression of FKBP5 increased the binding of SEC22B with SNAP29? These mechanistic studies are very important for revealing the exact role of FKBP5/SKA2-SA pathway in the regulation of neuroinflammation.

3. Fkbp5 KO mice had reduced IL-1 β secretion induced by acute stress. However, there is a lack of data to identify the effect is via FKBP5-SA pathway.

4. As we know, using AAV to transduce microglia efficiently is challenging. In this study, microglia had much lower transduction efficiency (Figure 2C) compared with neurons (Figure 2B) and astrocytes (Figure 2D). In this case, it is difficult to confirm the effect of Ska2 KD on IL-1 β release and neuroinflammation. As the key cell type focused by this study, a better method for efficient gene delivery to microglia should be applied.

5. As shown in figure 3J, the level of active inflammasome under control condition (without any stimulus) should be very low. Accordingly, there is a concern about the high ratio of ASC specks under control condition shown in figure 3A.

6. The authors successfully identified the hippocampal atrophy in the mice 6 weeks after viral-mediated KD of Ska2. What is the major mechanism of the cell death in this area, apoptosis, pyroptosis or necrosis?

Minor concerns:

The pictures of figure 2B (6 weeks Ska2-shRNA) and figure 2D (6 weeks Ska2-shRNA) are very similar. Please check and change to a better representative one.

Rebuttal letter to the reviewers' comments:

Reviewer #1 (Remarks to the Author):

In this paper by Hartmann entitled "SKA2 regulated hyperactive secretory autophagy drives neuroinflammation induced neurodegeneration" the authors present data to support a model where secretory autophagy (SA) regulates neuroinflammation-mediated neurodegeneration via SKA2 and FKBP5. The latter stimulates SA and release of interleukin 1beta (IL-1b) whereas SKA2 inhibits SA-dependent IL-1b release by counteracting FKBP5.

This is a very interesting paper with potentially very important findings. However, at the present stage I think that the conclusions are not completely validated by the data shown. Firstly, we need to be more convinced about how SKA2 and FKBP5 mechanistically act in SA. Secondly, is the dramatic effect of SKA2 knockdown (KD) on the hippocampus actually due to neuroinflammation caused by unchecked SA?

Major points:

1. Direct protein-protein interaction data are missing to strengthen the model of SKA2 and FKBP5 regulation of the SNARE complexes involved in secretory autophagy. Experiments testing out direct interactions and locating mutations that can compromise the interaction will strengthen this model. Does FKBP5 bind directly to SEC22B? Does SKA2 bind to STX3 or -4 or both or to SNAP29 or -23 or both? This needs to be tested.

We would like to thank the reviewer for raising these important questions. To address these concerns, we have conducted additional protein pull-down assays, now represented as **novel Fig. 1B and novel Fig. S1**, utilizing recombinant proteins. These assays serve to provide further validation of our IP/co-IP findings. Specifically, our investigations confirm a direct protein-protein interaction between SKA2 and SNAP29 (**novel Fig. 1B**). Interestingly, our experiments did not reveal any direct interactions between SKA2 and either SNAP23, STX3 or STX4. This suggests a potential interaction between SKA2 and these proteins via SNAP29 that are known to interact with each other as part of the SNARE-machinery. In a previous study¹, we identified FKBP5 as an interaction partner of the autophagosomal vSNARE-Protein SEC22B through an unbiased interactomics approach utilizing mass spectrometry. We extensively investigated the role of FKBP5 in secretory autophagy (SA). We now also conducted additional protein pull-down assays with recombinant FKBP5 and SEC22B. These results confirmed a direct protein-protein interaction between FKBP5 and SEC22B (**novel Fig. S1**). We included these novel findings in the results. While the identification of mutations affecting the SKA2-SNAP29 interaction remains an intriguing avenue for future exploration, we believe that this line of inquiry extends beyond the scope of the present study and should be addressed in forthcoming publications. We have expanded upon this matter in the 'Discussion' section (please also refer to the second to last paragraph of the discussion).

In summary, the novel findings from our direct protein-protein interaction data align with our initial results, thereby further strengthening the model of SKA2 and FKBP5 regulation of the SNARE complexes involved in secretory autophagy. Please also note that the results shown here were repeated in three independent experiments.

2. Another missing link is colocalization studies showing that SKA2 and FKBP5 colocalize with the relevant components in the SA complexes mentioned in 1.

The reviewer raises an important question, prompting us to conduct additional colocalization experiments for SKA2 and SNAP29 in human postmortem hippocampal tissue from control subjects now represented as **novel Fig. 7E-I**). Our results demonstrate that SKA2 co-localizes with SNAP29 on the cell surface and within the cytoplasm of neurons and glial cells, which aligns with our initial hypothesis and findings suggesting that SKA2 blocks SNARE-fusion at the plasma membrane by binding SNAP29. Confocal microscopy co-immunohistochemistry (IHC) images depict cells labeled with antibodies against SNAP29, SKA2 or DAPI, and the overlap of the three markers (E). Notably, the colocalization of SKA2 with SNAP29 was observed on the cell-surface and cytoplasm of neurons (F) as well as in microglia (G-I). High-resolution images capturing single slices through the z-axis indicate that the majority of SKA2 and SNAP29 labeling is not localized within the DAPI-positive nuclei (H&I). Considering our previous findings on the interaction of FKBP5 with SEC22B¹, the novel results of the protein pull-down experiments (**novel Fig. S1**), and the value of human postmortem tissue as a precious resource, we did not perform any additional co-IHC experiments involving FKBP5 and other relevant components in the SA complexes.

3. Fig 2A: Following viral-mediated shRNA-dependent KD of Ska2 in the hippocampus of C57Bl/6J mice complete hippocampal atrophy occurred within six weeks. As its name implies, spindle and kinetochore-associated complex subunit 2 (SKA2) has a known function as a scaffolding protein forming the SKA complex together with SKA1 and SKA3. This complex is a key component of the kinetochore-microtubule interface. The SKA complex is required for normal regulation of cell cycle checkpoint. KD of SKA2 expression by RNAi is known to block cell cycle progression during metaphase. Cells can often complete mitosis, but it is delayed, kinetochore fibers are destabilized and there is often a failure of spindle assembly checkpoint exit (EMBO J. 25, 5504, 2006). Mutations that disrupt the SKA complex can lead to cell death (Molecular Cell DOI 10.1016/j.molcel.2012.03.005).

The authors conclude from their KD SKA2 experiments in the hippocampus that: “Together, these data provide significant mechanistic evidence that hyperactivated SA (through KD of Ska2) is able to create an inflammatory feed-forward vicious cycle resulting in a GSDMD-mediated excessively neurotoxic environment to ultimately catalyze neuroinflammation and neurodegeneration.” The question is if this now occurs secondarily to cell death caused by SKA2 KD or if there is a direct effect on secretory autophagy of IL1b by removal of SKA2 as a negative regulator of SA? Hence, can the authors show that the effects they see is not due to apoptosis or other type of cell death? The neuroinflammatory effects may be secondary to the cell death.

We thank the reviewer for this insightful comment. We agree with the reviewer that the majority of neuroinflammatory effects observed at 2, 4 and 6 weeks are likely to be of a secondary nature. We sincerely apologize for any lack of clarity in the initial version of the manuscript regarding this aspect. In order to dissect this matter into more detail, we conducted additional experiments involving RNA sequencing 2 and 4 weeks following hippocampal knockdown of *Ska2* now represented as **novel Fig. 6 and novel Fig. S5**. These new findings further underscore the presence of multiple pathways and mechanisms, encompassing various functions of SKA2, which may contribute, either directly or indirectly, to the observed effects on cell death and neuroinflammation. We also refer the reviewer to our response to comment 6 from reviewer #2.

1. As pointed out by the reviewer, mutations disrupting the SKA complex can result in cell death in dividing cells (Molecular Cell DOI 10.1016/j.molcel.2012.03.005)², possibly contributing to the observed neurodegenerative phenotype observed at 2, 4 and 6 weeks. This crucial study has been

incorporated into our introduction and discussion, underscoring its significance (please also refer to the second to last paragraph of the introduction and the third paragraph of the discussion).

2. Another pathway influenced by SKA2 function, with neuroinflammation implications, is secretory autophagy, as demonstrated in our present study. Fig. 1 showcases associations of SKA2 with proteins (SNAP29, SNAP23, SEC22B, and STX3) involved in SA using IP/co-IP in mouse brain tissue, and highlights the direct SKA2-SNAP29 interaction in protein pull-down experiments using recombinant proteins (please also refer to our response to comment point 1, **novel Fig. 1B**). Together the SKA2-interaction experiments presents that SKA2-SNAP29 binding consequently results in a protein-complex with STX3 and SNAP23 in turn counteracting SEC22B-FKBP5-mediated autophagosome binding to the plasma membrane, which is functionally essential for secretory autophagy. In microglia cell cultures (SIM-A9 cells), SKA2 knockdown (KD), and overexpression of FKBP5 both enhance SNARE complex formation (SNAP29 binding to SEC22B and STX3 binding to SEC22B), indicative of increased SA activity. In addition, FKBP5 knockout enhances SKA2-SNAP29 binding, while FKBP5 overexpression yields the opposite effect. Furthermore, SKA2 overexpression diminishes FKBP5-SEC22B binding, whereas SKA2 KD increases it (Fig1 C-M). Hence, SKA2's influence on SNARE complex formation, important for autophagosome-plasma membrane fusion, can negatively regulate SA activity. Notably, many SA cargo proteins, including cytokines and cathepsins such as IL-1 β and Cathepsin D, are pivotal immune response factors with significant implications for neuroinflammation^{1,3}. Our data also demonstrates the impact of altered SKA2 expression on IL-1 β release, and as published previously of FKBP5's effects on Cathepsin D and IL-1 β release in microglia cell cultures (Figure 2A-B)¹. Moreover, our *in vivo* microdialysis findings establish the dependence of IL-1 β release in the PFC of mice on FKBP5 and the autophagy machinery. This is now further supported by our novel secretion experiments in microglia cultures using SAR405 (VPS34 inhibitor, VPS34i) and MRT68921 (ULK1 inhibitor, ULK1i), now represented as **novel Fig. S3** (please also refer to our response to comment 7). These collective results suggest that SKA2 can influence SA activity, subsequently impacting the release of established SA cargo proteins, including immune modulators driving neuroinflammation, as depicted in Fig. 2.
3. Heightened pro-inflammatory stimuli intensities, such as microbial components or endogenous cytokines, have been shown to trigger sequential activation of vesicular and Gasdermin D (GSDMD)-mediated IL-1 β secretory pathways⁴. Fig. 4 underscores how increased SA activity can induce inflammasome assembly and subsequent GSDMD-mediated IL-1 β release. Specifically, SKA2 depletion enhances SA-dependent IL-1 β release, creating a molecular feed-forward loop amplifying inflammasome activation. Inflammasome assembly activates Caspase-1 (CASP-1) enzymatic function. ASC1 as a component of the inflammasome complex recruits CASP-1. Activation of CASP-1 cleaves GSDMD to release the N-terminal domain, which forms pores in the plasma membrane for uncontrolled IL-1 β release.

To delve deeper into the mechanisms underpinning SKA2-mediated neuroinflammation/neurodegeneration, we employed RNA sequencing of hippocampal tissue 2 and 4 weeks post viral-mediated KD of Ska2, now represented as **novel Fig. 6 and novel Fig. S5**. We investigated whether transcriptional changes are associated with a hyperactivated SA pathway and subsequent inflammasome activation, apoptotic processes and cell death.

At the 2-week time point, differential expression analysis identified 3,479 differentially expressed genes (DEGs) ($p < 0.05$ adjusted with Bonferroni correction) of which 704 genes were upregulated and 21 downregulated with a Log₂ Fold Change (FC) > 2 (**novel Fig. 6A, left, Table S1**), while 5,562 genes were

differentially expressed 4 weeks after Ska2 KD (padj < 0.05), of which 505 genes were upregulated and 218 downregulated with a Log2FC > 2 (**novel Fig. 6B, left, Table S2**). We found that many of the DEGs are associated with an immune response. Interestingly, numerous upregulated genes belong to classes of immune mediators such as chemokines, cathepsins and cytokines including Il-1 β and Ctsd. Notably, several genes (**novel Fig. 6A-B, right**) including Il-1 β , Il-18, Ctsd, Ctsl, Ctsz, Ccl4 and Ccl5, have previously been shown to be released through SA^{1,3}. Thus, increased release of cargo proteins through SA from intracellular pools induced by KD of Ska2 might lead to increased mRNA expression and thus production of the secreted proteins. In fact, Ska2 KD resulted in a significant enrichment of genes encoding secreted proteins (Wilcoxon test, p < 0.0001, **novel Fig. 6C-D**) that have previously been identified in a secretome-wide analysis of SA in microglia cultures¹.

Next, we performed a Gene Ontology (GO) enrichment analysis which revealed numerous enriched terms of biological processes, cellular components and molecular functions at 2 and 4 weeks following Ska2 KD (**novel Fig. 6E-F**). For the entire list of enriched GO terms please refer to **novel Table S3 and S4**. Many of these GO terms such as 'cytokine-mediated signaling pathway', 'cytokine secretion', 'positive regulation of interleukin-1 beta secretion', 'positive regulation of interleukin-1 production', 'lysosome' and 'microglia activation' from the 2-week time point can be associated with increased activity of SA before substantial neuronal degeneration and cell death (**novel Fig. 6E**). In addition, the GO term 'inflammasome complex' was significantly enriched further supporting that cells are poised for inflammasome activation and subsequent increase in uncontrolled IL-1 β secretion via the GSDMD pathway. Similar GO terms were also enriched after 4 weeks (**novel Fig. 6F**). Notably, enriched GO terms such as 'extrinsic apoptotic signaling pathway' and 'positive regulation of cysteine-type endopeptidase activity involved in apoptotic process' also point towards an increased apoptotic activity at 2 and 4 weeks (only 'extrinsic apoptotic signaling pathway'). Moreover, the GO term 'condensed chromosome, centromeric region' was enriched at 2 weeks. This is in support of the hypothesis that there are multiple pathways/mechanisms involving various functions of SKA2 that may contribute, both directly and indirectly, to the observed neuroinflammatory effects. Interestingly, only at the 4-week time point, the GO enrichment analysis identified several enriched terms related to altered synaptic plasticity such as 'postsynaptic density', 'neuron to neuron synapse' and 'glutamatergic synapse' (**novel Fig. 6F**). This might relate to the progressive neuronal death observed at 4 weeks following Ska2 KD (Fig. 3A-B).

Next, we conducted a Kyoto Encyclopedia of Genes and Genomes (KEGG) pathway enrichment analysis with the DEGs. The results showed that the DEGs were highly associated with several immune pathways and neurodegenerative diseases, including 'cytokine-cytokine receptor interaction', 'NOD-like receptor signaling pathway', 'apoptosis', 'necroptosis', 'NF-kappa B signaling pathway', 'lysosome' and 'Alzheimer's disease' (**novel Fig. 6G-H, Table S5 and S6**).

The results from the differential expression analysis and GO and KEGG enrichment analyses indicate that a KD of Ska2 might have an impact on the cellular composition in the hippocampus. Therefore, we deconvoluted the bulk RNA-seq data with the multi-subject single-cell (MuSiC) method⁵. Confirming the IHC results (Fig. 3B and C), Ska2 KD led to altered estimated cell proportions in the hippocampus including decreased numbers of neurons and increased numbers of microglia (**novel Fig. S5A-B**; 2-way ANOVA: 2 weeks: condition x cell type interaction $F_{5,84} = 27.01$, p < 0.0001, Bonferroni's post hoc test: neuron, p < 0.001, microglia, p < 0.0001, astrocyte, p < 0.05; 4 weeks: condition x cell type interaction $F_{5,84} = 115.0$, p < 0.0001, Bonferroni's post hoc test: neuron, p < 0.001, microglia, p < 0.0001) at 2 and 4 weeks.

Together, the RNA sequencing analysis offers compelling additional support for the role of SKA2 as a significant negative regulator of SA within the brain. Furthermore, it unveils the intricate involvement of SKA2 in various functions spanning multiple pathways, which collectively contribute to neuroinflammation and the processes of neurodegeneration and cell death, both through direct and

indirect mechanisms. *We have now integrated these insights into both the results and discussion sections (please also refer to the results section “RNA sequencing analyses following hippocampal knockdown of Ska2 identify transcriptional signatures associated with increased activity of secretory autophagy and various cell death processes” and the 3rd paragraph of the discussion).*

Please note: The RNA sequencing data following Ska2 knockdown (Figure 6 and Fig. S5A-B) generated in this study are available at Gene Expression Omnibus (GEO): GSE181203.

The secure token to access the raw data can be requested from the editor. Accession codes will be available before publication.

4. The known roles of SKA2 are not presented in the paper at all which the authors certainly should as part of either the INTRODUCTION or DISCUSSION, or both. Can the authors rationalize some of the known roles with the role they suggest SKA2 has in secretory autophagy?

We would like to thank the reviewer for this important comment. We now included additional text in the introduction and discussion on the previously known roles of SKA2 such as its involvement in the SKA complex and its function in glucocorticoid receptor signaling and association with stress-related psychiatric disorders (please also refer to paragraph 4 of the introduction and paragraph 3 of the discussion).

5. Does SKA1 or SKA3 have any role in SA? Is SKA2 acting alone in SA without its partner proteins in the SKA complex?

We would like to thank the reviewer for raising these questions. Our additional analysis has demonstrated that SKA2 primarily functions independently of SKA1 and SKA3 in the context of SA (**novel Fig. S2**). Specifically, we conducted co-immunoprecipitation experiments following the procedures described in Fig. 1A. These experiments involved SNAP29-IP in tissue homogenates from the mouse hippocampus to investigate whether SKA1 or SKA3 associates with SNAP29, similar to SKA2. We suspect that this interaction with SNAP29 is the core mechanism underlying SA induced by SKA2. Our experimental findings suggest that there may be no interaction or association between SNAP29 and SKA1 or SKA3, thus providing support for the idea that SKA2 acts as the exclusive regulator of SA. We included these novel findings to the results (**novel Fig. S1**). Results shown here were repeated in three independent experiments.

6. Has the SKA2 Ab used for IP been validated for specificity? It would have been nice to see a negative control in the IP experiment in Fig. 1A where all tested proteins are positive.

We appreciate the reviewer for raising this question. The SKA2 antibody utilized for the IP experiments has been validated in mouse hippocampal tissue after KD using shRNA AAVs. We had already included this information in the initial submission of the manuscript, but we apologize if it was not explicitly clear. The results demonstrating the validation can be found in Fig. S4A.

In order to confirm SKA2 as interactor to plasma-membranous SNAREs while FKBP5 interacts to autophagosomal SNARE SEC22B (shown in Martinelli et al. 2021¹ and in new pull down experiments) we

now also provide evidence of FKBP5 as non-binding to SKA2. In the same SKA2-IP shown in Fig. 1A we also tested for FKBP5-association. Here we could not find any co-precipitation for FKBP5. This additionally proves that Ska2-IP technically was performed correctly.

7. The authors employ in Fig. 1R the ULK1 inhibitor (ULK1i) MRT68921 to show that autophagy is involved in the secretion of IL1beta. This experiment is very important and as it stands it rests on the specificity of this inhibitor for ULK1. The problem is that we cannot rule out TBK1 and AMPK which both are inhibited by MRT68921 and both of them are involved in autophagy (PMID: 25833948). The authors should also do this experiment using a VPS34 inhibitor like SAR405.

We would like to thank the reviewer for this comment. In response, we conducted additional experiments using SAR405 (VPS34 inhibitor, VPS34i) and MRT68921 (ULK1 inhibitor, ULK1i) in microglia cultures to validate our initial findings regarding the role of the autophagy machinery in the secretion of established SA cargo proteins, now represented in **novel Fig. S3**. Specifically, we evaluated Cathepsin D, a widely recognized SA cargo protein¹, by analyzing the supernatant of microglia cultures (SIM-A9 cells). This analysis aimed to assess Cathepsin D levels subsequent to SA induction via L-Leucyl-L-Leucine methyl ester (LLOMe) treatment, along with VPS34i or ULK1i treatment, respectively.

We have previously shown that Dex-induced release of Cathepsin D through SA is tightly linked to ATG5 function, a core protein of the autophagy machinery¹. Furthermore, we have established a clear role for FKBP5 in both macroautophagy and specifically SA, as demonstrated by the increase in early autophagy markers and autophagy flux in primary astrocytes overexpressing FKBP5⁶, as well as the absence of the DEX-induced Cathepsin D release in SIM-A9 *Fkbp5*-KO cells¹, respectively. These results already point to the importance of functional autophagy at various steps impacting the release of SA cargo proteins.

At the level of secreted Cathepsin D (**novel Fig. S3A-B**), co-treatment of cells with 0.25mM LLOMe and 1 μ M ULK1i already abolished the significant LLOMe-induced Cathepsin D release, which was further reduced to baseline levels with 10 μ M ULK1i. Interestingly, in cells co-treated with LLOMe and VPS34i, the inhibition of PIK3C3/Vps34 could only weakly reduce the levels of released Cathepsin D. It is possible that the additional inhibitory effect of ULK1i against AMPK and TBK1⁷ further diminishes SA, pointing towards the importance of functional autophagy regulation at different levels. Although both inhibitors influence the formation of autophagosomes, they might affect distinct subsets of proteins, leading to varied effects on SA. Furthermore, compensatory mechanisms have to be considered, as one inhibitor might trigger compensatory responses that affect SA differently than compensatory responses triggered by the other inhibitor.

To demonstrate the inhibitory activity of ULK1i and VPS34i against autophagy, we have performed autophagy flux measurements in the absence and presence of Bafilomycin A1 (BafA1) in SIM-A9 cell cultures. Both compounds led to a dose-dependent reduction of autophagy flux, evidenced by a decrease in LC3B-II protein levels, with VPS34i being more potent even at lower doses (0.1 and 1 μ M) compared to ULK1i (**novel Fig. S3C-D**).

Our results, based on pharmacological and genetic manipulations, highlight the extensive signalling crosstalk of autophagy pathways that impact SA and convincingly demonstrate the effectiveness of *Atg5*-KO¹, *Fkbp5*-KO¹, and ULK1 inhibition in influencing SA. This is in line with our initial hypothesis and *in vivo* microdialysis findings using the *Fkbp5* knockout mouse line as well as the ULK1 inhibitor in wild type mice.

Minor points:

8. *The blot in Fig 1H shows a band at the expected MW for FKBP5 in the extract from FKBP5 KO. Is the KO not complete, or is it the ab recognizing both FKBP51 and FKBP52 or another protein?*

We would like to thank the reviewer for this question. Indeed, we cannot rule out that the FKBP51 antibody we used also shows some affinity for homologous FKBP52. Therefore we blotted those lysates again and tested a new and highly specific antibody from cell signaling technologies. We are now able to present a blot without any background signal that might come from unspecific FKBP52-binding (Fig 1I).

9. *In The Abstract on line 49 the expression “The SKA2-mediated hyperactivation of SA” is not a correct expression as SKA2 is actually knocked down and as such not “mediating” it is rather the loss of SKA2 activity that mediates or leads to hyperactivation of SA.*

We would like to thank the reviewer for this comment. We agree with the reviewer and have adjusted the wording in the abstract accordingly.

10. *Line 139 “1” is missing in “MRT68921”*

We would like to thank the reviewer for this comment. We added the missing number accordingly.

Reviewer #2 (Remarks to the Author):

This study by Hartmann et al attempts to reveal the role of SKA2-secretory autophagy (SA) pathway in inflammation-mediated neurodegeneration. The authors identified that SKA2 and FKBP5 had the opposite effects on RQ-SNARE complex formation and the subsequent IL-1 β release. To further investigate the role of SKA2 in the regulation of neuroinflammation, AAV was injected into the mouse hippocampus to knock down Ska2, which results in microglia activation, inflammasome formation, hippocampal atrophy and cognitive impairment. Detection of postmortem brains from Alzheimer’s patients found the reduction of SKA2 level and increase of secretory autophagy (SA). Overall, it is an interesting study to identify a novel mechanism of regulating SA-dependent neuroinflammation. However, there are some major concerns about the study design and data analysis. The mechanistic studies should be further strengthened to support the conclusion.

Major concerns:

1. *Although the author used postmortem brain samples of AD patients, there is no evidence from any cellular or animal study to show the link of SKA2-SA pathway with any pathological hallmarks of AD, such as A β plaques, tau phosphorylation and neuronal death. Without these data, the role of SKA2 in the pathogenesis of AD cannot be confirmed.*

We would like to thank the reviewer for this insightful comment. In addition to well-known factors such as amyloid plaques, cerebral amyloid angiopathy, neurofibrillary tangles, neuronal and synaptic loss, neuroinflammation is increasingly recognized as a hallmark of neurodegenerative diseases including AD⁸⁻

¹³. This is supported by recent converging findings showing immune responses, including microglia activation, arising alongside degenerating neurons. While microglia functions are important for maintaining a healthy environment in the brain, accumulating evidence indicates that microglia may also negatively influence neuronal function and structure upon immune activation.

Given that our data suggest a role for SKA2 in SA and neuroinflammation-induced neurodegeneration, we further investigated whether an increased activity of the SA pathway is involved in the pathophysiology of AD. Our approach is comprehensive and translational, spanning from microglia cell cultures to mice models to human postmortem tissue from both control subjects and individuals with AD. Notably, the analysis of human postmortem AD tissue is particularly well suited to explore a potential link between SKA2-SA and the disease. Our findings reveal a decrease in SKA2 expression, and an increase in SNAP29 to SEC22B binding in two independent AD cohorts (Harvard Brain Tissue Resource Center (n=20), Manchester Brain Bank (n=77)), indicative of heightened SA activity in AD (Figure 7J-K).

Addressing the reviewer's suggestion, we also extended our investigation and conducted a KEGG pathway enrichment analysis with the differentially expressed genes (DEGs) following RNA sequencing of hippocampal tissue 2 weeks after viral mediated KD of Ska2 in C57/B6 mice. The results strongly associate the DEGs with 'Alzheimer's disease' (refer to **novel Fig. 6G, Table S5**). Notably, the RNA sequencing results indicate that a KD of Ska2 might have an impact on the cellular composition in the hippocampus. Therefore, we deconvoluted the bulk RNA-seq data with the multi-subject single-cell (MuSiC) method⁵. Confirming the IHC results (Fig. 3B and C), Ska2 KD led to altered estimated cell proportions in the hippocampus including decreased numbers of neurons (increased neuronal death) and increased numbers of microglia (**novel Fig. S5A-B**; 2-way ANOVA: 2 weeks: condition x cell type interaction $F_{5,84} = 27.01$, $p < 0.0001$, Bonferroni's post hoc test: neuron, $p < 0.001$, microglia, $p < 0.0001$, astrocyte, $p < 0.05$; 4 weeks: condition x cell type interaction $F_{5,84} = 115.0$, $p < 0.0001$, Bonferroni's post hoc test: neuron, $p < 0.001$, microglia, $p < 0.0001$) at 2 and 4 weeks. For more information regarding the RNA sequencing results, please also refer to our response to comment 3 of reviewer #1.

Additional support for a potential link between SA and its regulators, SKA2 and FKBP5, to AD is present in existing literature. Our previous secretome-wide analysis of SA cargo proteins in microglia cell cultures identified several AD-associated proteins including APP (amyloid beta precursor protein), VIM (vimentin) and LRP1 (LDLR-related protein 1)¹. Interestingly, SKA2 DNA methylation has been associated with a reduction in prefrontal cortex thickness¹⁴ and decreased SKA2 expression has been observed in surrounding tissue¹⁵. In addition, increased FKBP5 expression has been linked to AD in various brain regions, and higher FKBP5 levels have been associated with AD progression¹⁶. Notably, FKBP5 mRNA expression is increased in microglia of entorhinal cortex postmortem samples from individuals with AD¹⁷.

In summary, our current findings, along with the aforementioned studies, provide further substantiation for the involvement of the SA pathway and its key regulators, SKA2 and FKBP5, in AD pathology. It is worth mentioning that while we have focused on AD as an illustrative example, it may be plausible that SKA2, FKBP5, and the SA pathway may also play roles in other neurodegenerative diseases. We acknowledge this now in the limitations of the study in the second to last paragraph of the discussion.

2. The results of figure 1 show the interactions among SKA2, FKBP5 and the components of SNARE complex, but the underlying mechanisms are not clear. How do SKA2 and FKBP5 regulate each other (Figure 1B & H)? Which component of SNARE complex directly binds with SKA2? How does FKBP5 regulate the binding of SEC22B with SNAP29? It seems that FKBP5 regulates the expression of SEC22B (Figure 1E). Whether this

is the mechanism by which overexpression of FKBP5 increased the binding of SEC22B with SNAP29? These mechanistic studies are very important for revealing the exact role of FKBP5/SKA2-SA pathway in the regulation of neuroinflammation.

We would like to thank the reviewer for raising these important questions. To address these concerns, we have conducted additional protein pull-down assays, now represented as **novel Fig. 1B and Fig. S1**, utilizing recombinant proteins. These assays serve to provide further validation of our IP/co-IP findings. Specifically, our investigations confirm a direct protein-protein interaction between SKA2 and SNAP29. Interestingly, our experiments did not reveal any direct interactions between SKA2 and either SNAP23, STX3 or STX4. This suggests a potential interaction between SKA2 and these proteins via SNAP29 that are known to interact with each other as part of the SNARE-machinery. We also conducted additional protein pull-down assays with FKBP5 and SEC22B. These results confirmed a direct protein-protein interaction between FKBP5 and SEC22B. We included these novel findings in the revised results. Please also refer to our response to comment 1 of reviewer #1.

Based on the representative image (previous Fig. 1E, now Fig. 1F), it may appear that FKBP5 regulates the expression of SEC22B. However, this is not the case. We thoroughly examined all individual experimental replicates and found no significant quantitative differences to suggest that FKBP5 regulates the expression of SEC22B. To prevent any potential confusion, we have replaced the provided image with a more suitable representative.

3. Fkbp5 KO mice had reduced IL-1 β secretion induced by acute stress. However, there is a lack of data to identify the effect is via FKBP5-SA pathway.

We would like to thank the reviewer for this comment. In a previous study¹, we identified FKBP5 as an interaction partner of SEC22B through an unbiased approach utilizing mass spectrometry, which we now further confirmed with protein pull-down assays (please also refer to our response to comment 1 of reviewer #1, our response to comment 2 above, and **novel Fig. S1**). Notably, we extensively investigated the role of FKBP5 in stress-induced SA¹ which is in very strong support of the hypothesis that the reduced IL-1 β secretion in *Fkbp5* KO mice, induced by acute stress, is mediated via the FKBP5-SA pathway. Along these lines, we demonstrate in microglia cultures (SIMA-A9 cells) that overexpression of FKBP5 increases SEC22B to SNAP29 binding (indicative of increased SA activity; Fig 1F-G) as well as IL-1 β secretion (Fig. 2B).

In response to the reviewer's comment (and to comment 7 of reviewer#1), we conducted additional experiments using the autophagy inhibitors SAR405 (VPS34 inhibitor, VPS34i) and MRT68921 (ULK1 inhibitor, ULK1i) in microglia cultures to validate our initial findings regarding the role of the autophagy machinery in the secretion of established SA cargo proteins (represented in **novel Fig. S3**). Specifically, we evaluated Cathepsin D, a widely recognized SA cargo protein¹, by analyzing the supernatant of microglia cultures (SIM-A9 cells). This analysis aimed to assess Cathepsin D levels subsequent to SA induction via L-Leucyl-L-Leucine methyl ester (LLOMe) treatment, along with VPS34i or ULK1i treatment, respectively.

We have previously shown that Dex-induced release of Cathepsin D through SA is tightly linked to ATG5 function, a core protein of the autophagy machinery¹. Furthermore, we have established a clear role for FKBP5 in both macroautophagy and specifically SA, as demonstrated by the increase in early autophagy markers and autophagy flux in primary astrocytes overexpressing FKBP5⁶, as well as the absence of the DEX-induced Cathepsin D release in SIM-A9 *Fkbp5*-KO cells¹, respectively. These results already point to the importance of functional autophagy at various steps impacting the release of SA cargo proteins.

At the level of secreted Cathepsin D (**novel Fig. S3A-B**), co-treatment of cells with 0.25mM LLOMe and 1 μ M ULK1i already abolished the significant LLOMe-induced Cathepsin D release, which was further reduced to baseline levels with 10 μ M ULK1i. Interestingly, in cells co-treated with LLOMe and VPS34i, the inhibition of PIK3C3/Vps34 could only weakly reduce the levels of released Cathepsin D. It is possible that the additional inhibitory effect of ULK1i against AMPK and TBK1⁷ further diminishes SA, pointing towards the importance of functional autophagy regulation at different levels. Although both inhibitors influence the formation of autophagosomes, they might affect distinct subsets of proteins, leading to varied effects on SA. Furthermore, compensatory mechanisms have to be considered, as one inhibitor might trigger compensatory responses that affect SA differently than compensatory responses triggered by the other inhibitor.

To demonstrate the inhibitory activity of ULK1i and VPS34i against autophagy, we have performed autophagy flux measurements in the absence and presence of Bafilomycin A1 (BafA1) in SIM-A9 cell cultures. Both compounds led to a dose-dependent reduction of autophagy flux, evidenced by a decrease in LC3B-II protein levels, with VPS34i being more potent even at lower doses (0.1 and 1 μ M) compared to ULK1i (**novel Fig. S3C-D**).

Our results, based on pharmacological and genotype manipulations, highlight the extensive signalling crosstalk of autophagy pathways that impact SA and convincingly demonstrate the effectiveness of *Atg5*-KO¹, *Fkbp5*-KO¹, and ULK1 inhibition in influencing SA. This is in line with our initial hypothesis and *in vivo* microdialysis findings using the *Fkbp5* knockout mouse line as well as the ULK1 inhibitor in wild type mice.

4. As we know, using AAV to transduce microglia efficiently is challenging. In this study, microglia had much lower transduction efficiency (Figure 2C) compared with neurons (Figure 2B) and astrocytes (Figure 2D). In this case, it is difficult to confirm the effect of Ska2 KD on IL-1 β release and neuroinflammation. As the key cell type focused by this study, a better method for efficient gene delivery to microglia should be applied.

We thank the reviewer for this insightful comment. We agree with the reviewer that AAV transduction rates of microglia *in vivo* are generally known to be low¹⁸. Currently, there are no cell type-specific SKA2 knockout mouse lines available to further dissect the role of SKA2 in microglia *in vivo*. However, we believe that our results in microglia cell cultures as well as the co-immunohistochemistry data of human microglia is in strong support of SKA2's role on IL-1 β release and neuroinflammation. Nonetheless, we included this limitation of the study to the discussion (please also refer to the second to last paragraph of the discussion) and suggest that future studies will be needed to further dissect the role of SKA2 in SA and neuroinflammation in a cell type-specific manner.

5. As shown in figure 3J, the level of active inflammasome under control condition (without any stimulus) should be very low. Accordingly, there is a concern about the high ratio of ASC specks under control condition shown in figure 3A.

We thank the reviewer for highlighting the observation of relatively high inflammasome activity under control conditions (without any stimulus) in previous Fig. 3A, now Fig. 4A.

In this manuscript, we initially outline the generation of SIM-A9 cells, which are spontaneously immortalized mouse microglia, engineered to stably express ASC-mCerulean, providing a visual indicator

of inflammasome activation. The elevated level of active inflammasome in the absence of any specific stimulus may potentially be attributed to the overexpression of the ASC reporter protein itself, potentially promoting self-association and speck formation. Moreover, it's worth noting that the propensity for ASC speck formation may vary depending on the specific cell type.

However, a more plausible explanation for the observed inflammasome activation may be attributed to cellular stress induced by the transfection procedure and the delivery of shRNAs, which could activate stress pathways, including those associated with inflammation.

We hope that the reviewer continues to consider this data as supportive evidence for the findings presented in Figure 4, which pertain to inflammasome activation following Ska2 knockdown *in vivo*.

6. The authors successfully identified the hippocampal atrophy in the mice 6 weeks after viral-mediated KD of Ska2. What is the major mechanism of the cell death in this area, apoptosis, pyroptosis or necrosis?

We appreciate the reviewer for raising this insightful question. We performed RNA sequencing analyses 2 and 4 weeks following hippocampal Ska2 knockdown to further investigate the pathways that led to the observed hippocampal atrophy (now represented as **novel Fig. 6 and Fig. S5**). Apoptosis, pyroptosis, and necroptosis are three different forms of programmed cell death. While each form has distinct signaling pathways and outcomes, there is also cross-talk and overlap in the molecular components involved and they share some common genes and pathways in their regulation. Our initial data (Fig. 4) and the RNA sequencing analysis (**novel Fig. 6 and Fig. S5**) suggest that Ska2 KD induces, directly and indirectly (via increased SA activity), multiple mechanisms of cell death including apoptosis, necroptosis and pyroptosis, in parallel. Please also refer to our response to comment 3 of reviewer #1.

Apoptosis, is a controlled and regulated process that eliminates unwanted or damaged cells without causing inflammation¹⁹. During apoptosis, cells undergo characteristic morphological changes, including cell shrinkage, chromatin condensation, nuclear fragmentation, and the formation of membrane-bound apoptotic bodies. Apoptosis is mediated by a family of proteins known as caspases, which are activated in response to various internal and external signals. The RNA sequencing analyses (**novel Fig. 6 and Fig. S5**) identified molecular signatures related to apoptosis. In particular, the Gene Ontology (GO) enrichment analysis revealed numerous enriched terms such as 'extrinsic apoptotic signaling pathway', 'positive regulation of cysteine-type endopeptidase activity involved in apoptotic process' and 'condensed chromosome, centromeric region' that point towards an increased apoptotic activity following Ska2 KD. Moreover, the Kyoto Encyclopedia of Genes and Genomes (KEGG) pathway enrichment analysis with the DEGs identified 'apoptosis' at the 2 week timepoint following Ska2 KD.

Necroptosis is a programmed form of necrosis, a type of cell death that is considered uncontrolled and inflammatory²⁰. Necroptosis shares some similarities with apoptosis in that it is also a regulated process, but it leads to cell rupture and release of cellular contents, which can trigger inflammation in the surrounding tissues. Necroptosis is mediated by a complex of proteins including receptor-interacting protein kinases and mixed lineage kinase domain-like protein. It can be triggered when certain death receptors such as Tumor Necrosis Factor Receptor 1 (TNFR1) are activated. Notably, TNFR1 can initiate both apoptosis and necroptosis pathways. Indeed, the KEGG pathway enrichment analysis with the DEGs not only identified 'apoptosis' but also 'necroptosis' as well as 'TNF signaling pathway' at the 2 week timepoint following Ska2 KD.

Pyroptosis is another highly inflammatory form of programmed cell death^{21,22}. Upon detection of danger signals (including endogenous cytokines such as IL-1 β), the inflammasome sensors such as NLRP3 (NOD-,

LRR- and pyrin domain-containing protein 3) form aggregates called ASC (apoptosis-associated speck-like protein containing a CARD) specks. These specks act as a platform to recruit and organize additional components of the inflammasome. Activated inflammasomes recruit and activate Caspase-1 (CASP-1). CASP-1 cleaves Gasdermin-D (GSDMD). The N-terminal fragment of GSDMD forms pores in the plasma membrane of the cell. This allows ions and water to flow into the cell, causing osmotic swelling and cell membrane rupture. The rupture of the cell membrane leads to the uncontrolled release of intracellular contents, including pro-inflammatory cytokines such as IL-1 β and IL-18. The release of inflammatory molecules promotes inflammation and attracts more immune cells to the site. Increasing intensities of pro-inflammatory stimuli have been shown to induce sequential activation of vesicular and GSDMD-mediated IL-1 β secretory pathways⁴. Our initial data in Fig. 4 and the RNAseq data (**novel Fig. 6 and Fig. S5**) suggest that increased activity of secretory autophagy is able to induce inflammasome formation and subsequent GSDMD-mediated IL-1 β release and, thus that SA is able to contribute to the initiation of pyroptosis. In particular, we demonstrate that increased activity of SA increases the number of ASC specks in microglia cell cultures and following hippocampal Ska2 KD in mice. In addition, KD of Ska2 resulted in increased binding of SEC22B to SNAP29, reflective of enhanced SA activity, along with increased inflammasome activation (NEK7 binding to NLRP3), in protein lysates of organotypic hippocampal slice cultures. CASP-1 expression was increased following hippocampal Ska2 KD, indicative of inflammasome activation. Notably, full length (FL) GSDMD as well as its cleaved N-terminal domain (GSDMD N-term) were increased at 2 weeks following KD of hippocampal Ska2. Along these lines, the GO analysis of the RNAseq data identified the term 'inflammasome complex' to be enriched at 2 weeks. Together, the data suggest that pyroptosis (via increased activity of SA) is involved in the observed hippocampal atrophy. In addition, it may be possible that Ska2 might also have a direct effect on pyroptosis, independent from its function in SA.

Collectively, the data suggest that following Ska2 KD multiple mechanisms of cell death including apoptosis, necroptosis and pyroptosis, lead to the observed hippocampal atrophy. While these three forms of cell death have distinct features, the underlying molecular pathways are not entirely separate and can interact in intricate ways. We now included these findings to the results and discussion.

Our data (Fig. 4) along with the RNA sequencing analysis (**novel Fig. 6 and Fig. S5**) indicate that Ska2 KD triggers various pathways of cell death, encompassing apoptosis, necroptosis, and pyroptosis, both directly and indirectly (through heightened SA activity), simultaneously. We have now included this in the results (RNAseq section) and discussion (paragraph 3).

Minor concerns:

The pictures of figure 2B (6 weeks Ska2-shRNA) and figure 2D (6 weeks Ska2-shRNA) are very similar. Please check and change to a better representative one.

We thank the reviewer for this comment and we apologize for the mix up in previous Fig. 2B (now Fig. 3B). We replaced the image with the correct representative.

References

1. Martinelli, S. *et al.* Stress-primed secretory autophagy promotes extracellular BDNF maturation by enhancing MMP9 secretion. *Nat. Commun.* 2021 121 **12**, 1–17 (2021).
2. Jeyapragash, A. A. *et al.* Article Structural and Functional Organization of the Ska Complex, a Key

- Component of the Kinetochore-Microtubule Interface. *Mol. Cell* **46**, 274–286 (2012).
3. Kimura, T. *et al.* Dedicated SNARE s and specialized TRIM cargo receptors mediate secretory autophagy . *EMBO J.* **36**, 42–60 (2017).
 4. Semino, C., Carta, S., Gattorno, M., Sitia, R. & Rubartelli, A. Progressive waves of IL-1 β release by primary human monocytes via sequential activation of vesicular and gasdermin D-mediated secretory pathways. *Cell Death Dis.* **9**, 1088 (2018).
 5. Wang, X., Park, J., Susztak, K., Zhang, N. R. & Li, M. Bulk tissue cell type deconvolution with multi-subject single-cell expression reference. *Nat. Commun.* **10**, 1–9 (2019).
 6. Gassen, N. C. *et al.* Association of FKBP51 with Priming of Autophagy Pathways and Mediation of Antidepressant Treatment Response: Evidence in Cells, Mice, and Humans. *PLoS Med.* **11**, (2014).
 7. Petherick, K. J. *et al.* Pharmacological Inhibition of ULK1 Kinase Blocks Mammalian Target of Rapamycin (mTOR)-dependent Autophagy * □ S. (2015) doi:10.1074/jbc.C114.627778.
 8. Heneka, M. T. *et al.* Neuroinflammation in Alzheimer’s disease. *The Lancet Neurology* vol. 14 388–405 (2015).
 9. Ising, C. *et al.* NLRP3 inflammasome activation drives tau pathology. *Nature* **575**, 669–673 (2019).
 10. Heneka, M. T., McManus, R. M. & Latz, E. Inflammasome signalling in brain function and neurodegenerative disease. *Nature Reviews Neuroscience* vol. 19 610–621 (2018).
 11. Ising, C. & Heneka, M. T. Functional and structural damage of neurons by innate immune mechanisms during neurodegeneration review-Article. *Cell Death and Disease* vol. 9 (2018).
 12. Heneka, M., Kummer, M. & Latz, E. Innate immune activation in neurodegenerative disease. *Nat. Rev. Immunol.* **14**, 463–477 (2014).
 13. Venegas, C. *et al.* Microglia-derived ASC specks crossseed amyloid- β in Alzheimer’s disease. *Nature* **552**, 355–361 (2017).
 14. Sadeh, N. *et al.* SKA2 methylation is associated with decreased prefrontal cortical thickness and greater PTSD severity among trauma-exposed veterans. *Mol. Psychiatry* **21**, 357–363 (2016).
 15. Guintivano, J. *et al.* Identification and replication of a combined epigenetic and genetic biomarker predicting suicide and suicidal behaviors. *Am. J. Psychiatry* **171**, 1287–1296 (2014).
 16. Blair, L. J. *et al.* Accelerated neurodegeneration through chaperone-mediated oligomerization of tau. *J. Clin. Invest.* **123**, (2013).
 17. Grubman, A. *et al.* A single-cell atlas of entorhinal cortex from individuals with Alzheimer’s disease reveals cell-type-specific gene expression regulation. *Nat. Neurosci.* **22**, 2087–2097 (2019).
 18. Maes, M. E., Colombo, G., Schulz, R. & Siegert, S. Targeting microglia with lentivirus and AAV: Recent advances and remaining challenges. *Neurosci. Lett.* **707**, 134310 (2019).
 19. Taylor, R. C., Cullen, S. P. & Martin, S. J. Apoptosis: controlled demolition at the cellular level. *Nat. Rev. Mol. Cell Biol.* 2008 93 **9**, 231–241 (2008).
 20. Vandenabeele, P., Galluzzi, L., Vanden Berghe, T. & Kroemer, G. Molecular mechanisms of necroptosis: an ordered cellular explosion. *Nat. Rev. Mol. Cell Biol.* 2010 1110 **11**, 700–714 (2010).
 21. Liu, X. *et al.* Inflammasome-activated gasdermin D causes pyroptosis by forming membrane pores. *Nature* **535**, 153–158 (2016).
 22. Kesavardhana, S., Malireddi, R. K. S. & Kanneganti, T.-D. Caspases in Cell Death, Inflammation, and Pyroptosis. *Annu. Rev. Immunol.* **38**, 567–595 (2020).

REVIEWER COMMENTS

Reviewer #1 (Remarks to the Author):

The authors have answered all my questions, addressed the points of critique and suggestions for improvement I made, and revised the manuscript in a satisfactory manner.

Reviewer #2 (Remarks to the Author):

The authors performed additional experiments and the revised manuscript has addressed the reviewer's comments.

Reviewer #3 (Remarks to the Author):

I was specifically asked to review the transcriptomic analyses added to this study. In this paper, Hartmann et al. introduced bulk RNA-seq data to identify the differentially expressed genes (DEGs) between the hippocampi of male Ska2-KD and control mice at 2 and 4 weeks to determine if hyperactive secretory autophagy (SA) via Ska2-KD promotes neuroinflammation through direct triggering of molecular pathways in addition to the indirect contribution by Ska2-KD induced cell death. The authors identify DEGs at both 2 and 4 weeks pointing to many pathways linked to neuroinflammation. They then perform a deconvolution analysis and confirm altered cell type proportions.

The overarching story of this paper is compelling and the addition of the RNA-seq analysis strengthens the study and works to disentangle the direct versus indirect effects of SA on neuroinflammation. However, the authors stopped a step short in confirming the transcriptional changes are directly associated with Ska2-KD, rather than the cell type proportion changes they identified in the deconvolution analysis.

1. Gratefully, after the authors identified DEGs and GO/KEGG pathways linked to neuroinflammation, they also recognized that these same findings could point to differential proportions of cell types and therefore, performed a deconvolution analysis. They confirmed decreased neuronal and increased microglial proportions. Unfortunately, this discovery calls the previously made conclusions from the DE analysis and enrichment analyses into question. Are the DEGs and subsequent GO/KEGG pathways and secreted protein enrichments driven by these cell-type proportion changes? Microglia (and microglia expressed genes) are associated with inflammation and there is a large increase in the proportions of microglia. Additionally, the secreted-protein genes were specifically identified in an analysis of microglia. Therefore, the identified DEGs could simply be reflecting these proportion changes rather than any true

transcriptional changes driven by Ska2-KD. At a minimum, these cell-type proportions should be accounted for in the model when identifying DEGs. Additional data from sorted microglia or single-cell RNA-seq should be conclusive, but potentially beyond the scope of this study.

Minor

1. MuSiC was used to deconvolve the bulk RNA into cell-type proportions. Technically, MuSiC2 would have been more appropriate for this analysis because it is designed to account for differential expression of marker genes between a healthy single cell reference data and a 'disease' bulk query. I'm considering this minor because the results shouldn't be terribly different, but using MuSiC2 would inspire more confidence in the readers. Additionally, it would have identified cell-type-specific DEGs at the same time.
2. Make it easier to find the number of biological replicates used in the RNA-seq analysis. Shift "n=8 per group" from (H) to (B) in the Figure 6 legend and/or add it to the methods. This is an impressive number of replicates, don't hide it.
 - a. "BP, biological process, CC, cellular component" should be moved to (F)

Rebuttal letter to the reviewers' comments:

Reviewer #1 (Remarks to the Author):

The authors have answered all my questions, addressed the points of critique and suggestions for improvement I made, and revised the manuscript in a satisfactory manner.

We are grateful to the reviewer for their time and effort. We are pleased to have addressed all their questions satisfactorily.

Reviewer #2 (Remarks to the Author):

The authors performed additional experiments and the revised manuscript has addressed the reviewer's comments.

We are thankful to the reviewer for their time and effort. We are glad to have successfully responded to all of their comments.

Reviewer #3 (Remarks to the Author):

I was specifically asked to review the transcriptomic analyses added to this study. In this paper, Hartmann et al. introduced bulk RNA-seq data to identify the differentially expressed genes (DEGs) between the hippocampi of male Ska2-KD and control mice at 2 and 4 weeks to determine if hyperactive secretory autophagy (SA) via Ska2-KD promotes neuroinflammation through direct triggering of molecular pathways in addition to the indirect contribution by Ska2-KD induced cell death. The authors identify DEGs at both 2 and 4 weeks pointing to many pathways linked to neuroinflammation. They then perform a deconvolution analysis and confirm altered cell type proportions.

The overarching story of this paper is compelling and the addition of the RNA-seq analysis strengthens the study and works to disentangle the direct versus indirect effects of SA on neuroinflammation. However, the authors stopped a step short in confirming the transcriptional changes are directly associated with Ska2-KD, rather than the cell type proportion changes they identified in the deconvolution analysis.

1. Gratefully, after the authors identified DEGs and GO/KEGG pathways linked to neuroinflammation, they also recognized that these same findings could point to differential proportions of cell types and therefore, performed a deconvolution analysis. They confirmed decreased neuronal and increased microglial proportions. Unfortunately, this discovery calls the previously made conclusions from the DE analysis and enrichment analyses into question. Are the DEGs and subsequent GO/KEGG pathways and secreted protein enrichments driven by these cell-type proportion changes? Microglia (and microglia expressed genes) are associated with inflammation and there is a large increase in the proportions of microglia. Additionally, the secreted-protein genes were specifically identified in an analysis of microglia. Therefore, the identified DEGs could simply be reflecting these proportion changes rather than any true transcriptional changes driven by Ska2-KD. At a minimum, these cell-type proportions should be accounted for in the model when identifying DEGs. Additional data from sorted microglia or single-cell RNA-seq should be conclusive, but potentially beyond the scope of this study.

We are thankful to the reviewer for their time and effort. We would like to express our gratitude to the reviewer for highlighting these crucial questions. In response to these concerns, we have normalized the Differentially Expressed Genes (DEGs) and subsequent analyses based on the estimated cell type proportions using the multi-subject single-cell (MuSiC) method. Additionally, we refer the reviewer to our detailed response below regarding the software package MuSiC2. We have relocated the figure illustrating the estimated cell proportions from the supplemental material (previously Figure S5A-B) to the main body of the manuscript (now Figure 6A-B). Given that the changes in estimated cell proportions showed a high correlation across different cell types (as illustrated in the **new Figure S5**), we normalized the bulk RNA-seq data based on the variations in the proportion of microglia. This approach accounts for changes in cell types.

The adjustment for cell type resulted in minor alterations in the DEGs at both the 2-week and 4-week marks, as illustrated in the following Venn diagrams (not included in this manuscript):

Furthermore, this deconvolution led to slight modifications in the outcomes of the subsequent enrichment analyses compared to the initial findings. In summary, the overarching conclusion drawn from the data remains consistent. The refined RNA-seq findings further emphasize the involvement of multiple pathways and mechanisms, including secretory autophagy, in the diverse functions of SKA2. These pathways and mechanisms may contribute, either directly or indirectly, to the observed effects on cell death and neuroinflammation.

For additional details, please refer to the updated RNA-seq section of the manuscript.

Minor

1. MuSiC was used to deconvolve the bulk RNA into cell-type proportions. Technically, MuSiC2 would have been more appropriate for this analysis because it is designed to account for differential expression of marker genes between a healthy single cell reference data and a 'disease' bulk query. I'm considering this minor because the results shouldn't be terribly different, but using MuSiC2 would inspire more confidence in the readers. Additionally, it would have identified cell-type-specific DEGs at the same time.

We appreciate the reviewer's comment and have taken it into consideration. In response, we initially attempted to deconvolve the bulk RNA-seq data into cell type proportions using the MuSiC2 software package. However, we encountered several challenges with this package, primarily due to what seems to be inadequate maintenance on GitHub. Despite multiple attempts, these issues remained unresolved, prompting us to opt for an alternative approach. This alternative involved the identification of Differentially Expressed Genes (DEGs) based on normalized estimated cell proportions using the original

MuSiC method. For additional information, please refer to our detailed response to your earlier comment.

2. Make it easier to find the number of biological replicates used in the RNA-seq analysis. Shift “n=8 per group” from (H) to (B) in the Figure 6 legend and/or add it to the methods. This is an impressive number of replicates, don’t hide it. a. “BP, biological process, CC, cellular component” should be moved to (F)

We would like to thank the reviewer for these comments which we implemented accordingly.

REVIEWER COMMENTS

Reviewer #3 (Remarks to the Author):

I want to highlight a potential error in implementing DESeq2 with your updated model, which included the microglia proportion as a covariate. Most of the genes have identical log2FCs, p-values, and padj-values as your previous results, indicating that microglia proportion did not get correctly incorporated into the model. The differences you saw in your Venn diagram suggesting that the model had run correctly are likely due to the updated DESeq2 version you used, which might have slightly different default thresholds. For instance, the first difference is in the 398th most significant gene in the new DEG results, ENSMUSG00000047562. This gene has a log2FC of 14.856, which is extreme and potentially removed by different default thresholds in the older version of DESeq2 that you used originally. A brief scan of 'new' hits showed that most fall into this category of extreme log2FCs. I suggest checking that microglia proportion was correctly added to the model and confirming that the thresholds for DESeq2 are what you expected.

Unfortunately, your github with the code is not publicly available yet, so I could not investigate to identify the exact location of the error for you. This error needs to be corrected along with updating the downstream GO and KEGG analyses.

Reviewer #3 (Remarks on code availability):

The URL did not work. Probably because it is not set public yet.

Reviewer #3 (Remarks to the Author):

I want to highlight a potential error in implementing DESeq2 with your updated model, which included the microglia proportion as a covariate. Most of the genes have identical log2FCs, p-values, and padj-values as your previous results, indicating that microglia proportion did not get correctly incorporated into the model. The differences you saw in your Venn diagram suggesting that the model had run correctly are likely due to the updated DESeq2 version you used, which might have slightly different default thresholds. For instance, the first difference is in the 398th most significant gene in the new DEG results, ENSMUSG00000047562. This gene has a log2FC of 14.856, which is extreme and potentially removed by different default thresholds in the older version of DESeq2 that you used originally. A brief scan of 'new' hits showed that most fall into this category of extreme log2FCs. I suggest checking that microglia proportion was correctly added to the model and confirming that the thresholds for DESeq2 are what you expected.

Unfortunately, your github with the code is not publicly available yet, so I could not investigate to identify the exact location of the error for you. This error needs to be corrected along with updating the downstream GO and KEGG analyses.

Dear Reviewer #3,

We would like to express our sincere gratitude for highlighting this error in implementing DESeq2 with the updated model. Indeed, we made a mistake when labeling the cell type proportions derived from MuSiC, which led to an erroneous use of the proportions. We adapted our scripts accordingly [code line 69 until 71 in the RNAseq_analysis.R script] and made the repository accessible on GitHub. The original data has already been stored at GEO (*see below*). We apologize that the code had previously not been publicly available.

The results of the updated RNA-seq analyses have not altered the overall conclusions and still provide compelling evidence that SKA2 acts a critical negative regulator of the SA pathway in the brain. The results support the hypothesis that hyperactivation of the SA pathway (triggered by Ska2 KD) initiates an inflammatory feed-forward loop which, based on the revised analyses, is especially notable at the 2-week time point. Specific details are provided below as well as in the RNA-seq section of the revised manuscript.

Correlation analyses across the phenotype of interest and relevant covariates, including cell type proportions now show that the *Ska2* knockdown at 2 and 4 weeks -as well as most cell type proportions derived from MuSiC- map on the first principal component of the data, suggesting a strong overall influence of the knockdown and the cell type proportions on the data (**Figure S5** of the manuscript). In addition, and now correctly depicted, the knockdown variable is highly correlated with most cell type proportions, suggesting that correcting for cell type composition is critical to investigate the underlying transcriptional mechanisms beyond cell type changes. The correlation plot also shows that most cell types are intercorrelated (**Figure S5** of the manuscript).

Figure S5. Correlation analysis of the variable of interest and relevant covariates.

Based on the correlation analyses, we implemented 4 different models for utmost transparency. However, we suggest presenting the model adjusted for microglia cell type proportions only in the final version of the manuscript.

Model 1 (prior model without adjustment for cell type proportions, as reference)

This model resulted in n=3482 FDR-significant DEGs for the week 2 comparison and n=5573 FDR-significant DEGs for the week 4 comparison (*DEGs listed in the attached Excel spreadsheets: DEG_analysis_week2_V2 and DEG_analysis_week4_V2*). However, as discussed before, the majority of the DEGs may be due to changing cell type proportions after *Ska2* knockdown.

Model 2 (adjusting for microglia (MG) cell type proportions)

As suggested by the reviewer, outlined above, and shown in **Figure S5**, most cell type proportions are highly correlated with the proportion of microglia (MG) in the data. Therefore, adjusting for MG can represent the other cell type proportions in the model. This model led to a reduced number of FDR-significant DEGs ($n=1367$ at week 2 and $n=601$ at week 4, respectively; *DEGs listed in the attached Excel spreadsheets: DEG_analysis_week2_V2 and DEG_analysis_week4_V2*). Interestingly, the subsequent GO analysis for the week 2-time point shows strong enrichment of GO terms related to immune system activation and inflammation, which we propose as an underlying mechanism of *Ska2* knockdown. In contrast, at week 4, we could only detect 3 significantly enriched GO terms (*GO analysis results listed in the attached Excel spreadsheets: enrichment_analysis_week2_V2 and enrichment_analysis_week4_V2*).

Model 3 (adjusting for MG and mural cell proportions)

As shown in **Figure S5**, mural cell type proportions are not correlated with the proportion of MG cells in the data. Adjusting for MG *and* mural cells may capture additional variation in the data. However, because mural cell type proportions are also correlated with almost all other cell types, this model may already lead to some degree of overfitting. We found n=1012 FDR-significant DEGs at 2 weeks and n=646 FDR-significant DEGs at 4 weeks, respectively (*DEGs listed in the attached Excel spreadsheets: DEG_analysis_week2_V2 and DEG_analysis_week4_V2*). Similar to the observation with Model 2, we detected a large number of GO terms related to immune system activation and inflammation enriched at week 2 but not at week 4 (*GO analysis results listed in the attached Excel spreadsheets: enrichment_analysis_week2_V2 and enrichment_analysis_week4_V2*)

Model 4 (adjusting for all cell type proportions)

We also implemented a model including all cell types. Given the observation that most cell types are highly intercorrelated, this model risks overfitting and is therefore presented as an exploratory analysis for the reviewer only. Surprisingly, we still find n=256 FDR-significant DEGs in week 2 and n=78 in week 4 (*DEGs listed in the attached Excel spreadsheets: DEG_analysis_week2_V2 and DEG_analysis_week4_V2*). In addition, we found n=166 enriched GO terms in week 2 but none in week 4 (*GO analysis results listed in the attached Excel spreadsheets: enrichment_analysis_week2_V2 and enrichment_analysis_week4_V2*).

Overlap in DEGs and enriched GO terms at 2 and 4 weeks.

All models showed a similar signature of DEGs and enriched terms, as illustrated in the Venn diagrams (not included in the manuscript). Given the collinearity of the cell types, we suggest presenting the model, including the MG cell type proportions, in the final version of the manuscript.

Our GitHub with the code is now publicly available (<https://github.com/klengellab/Ska2>). Additionally, we would like to extend the key to the submitted data to the reviewer with the token **ojsjmwsmrkbbcf** to access **GSE181203**.

REVIEWERS' COMMENTS

Reviewer #3 (Remarks to the Author):

The authors have corrected the differential expression analysis by microglia proportion and the revised manuscript appears to be in order.

Reviewer #3 (Remarks on code availability):

The code includes a README and detailed comments that should be satisfactory for reproducibility.